# PhysLLM: Harnessing Large Language Models for Cross-Modal Remote Physiological Sensing

**Yiping Xie**[1,2*], **Bo Zhao**[2*], **Mingtong Dai**[2], **Jian-Ping Zhou**[5,6], **Yue Sun**[7], **Tao Tan**[7],
**Weicheng Xie**[1†], **Linlin Shen**[1,3] **& Zitong Yu**[2,3,4†]

[1]Shenzhen University    [2]Great Bay University
[3]National Engineering Laboratory for Big Data System Computing Technology
[4]Dongguan Key Laboratory for Intelligence and Information Technology
[5]Guangdong Medical University    [6]Southern Medical University (Dongguan People's Hospital)
[7]Macao Polytechnic University

```
2310275054@email.szu.edu.cn, 225040248@link.cuhk.edu.cn
mt.dai@siat.ac.cn, zhoujianping0165@smu.edu.cn
yuesun@mpu.edu.mo, taotanjs@gmail.com
wcxie@szu.edu.cn, llshen@szu.edu.cn, zitong.yu@ieee.org
```

## Abstract

Remote photoplethysmography (rPPG) enables non-contact physiological measurement but remains highly susceptible to illumination changes, motion artifacts, and limited temporal modeling. Large Language Models (LLMs) excel at capturing long-range dependencies, offering a potential solution but struggle with the continuous, noise-sensitive nature of rPPG signals due to their text-centric design. To bridge this gap, we introduce the PhysLLM, a collaborative optimization framework that synergizes LLMs with domain-specific rPPG components. Specifically, the Text Prototype Guidance (TPG) strategy is proposed to establish cross-modal alignment by projecting hemodynamic features into LLM-interpretable semantic space, effectively bridging the representational gap between physiological signals and linguistic tokens. Besides, a novel Dual-Domain Stationary (DDS) Algorithm is proposed for resolving signal instability through adaptive time-frequency domain feature re-weighting. Finally, rPPG task-specific cues systematically inject physiological priors through physiological statistics, environmental contextual answering, and task description, leveraging cross-modal learning to integrate both visual and textual information, enabling dynamic adaptation to challenging scenarios like variable illumination and subject movements. Evaluation on four benchmark datasets, PhysLLM achieves state-of-the-art accuracy and robustness, demonstrating superior generalization across lighting variations and motion scenarios.

## 1 Introduction

Remote photoplethysmography (rPPG) is a non-contact technique that allows for the remote measurement of physiological signals such as heart rate (HR) (Yue et al., 2021; Das et al., 2023) and blood pressure (Wu et al., 2022) by analyzing subtle color changes in the skin caused by blood flow. Unlike traditional contact-based methods, such as electrocardiograms (ECG) and Photoplethysmograph (PPG) (Murthy et al., 2015), rPPG does not require physical sensors attached to the body, making it a more convenient and less intrusive option for continuous health monitoring. Based on these advantages of rPPG, more and more excellent work and research have been committed to improving its accuracy, robustness and scalability in recent years.

Traditional remote photoplethysmography (rPPG) methods (Poh et al., 2010; Madej et al., 2011; De Haan & Jeanne, 2013; Wang et al., 2017) typically rely on signal processing techniques to isolate rPPG signals from videos. More recent deep learning methods, including CNNs (Yu et al., 2019;

---

*Equal contribution
†Corresponding authors.

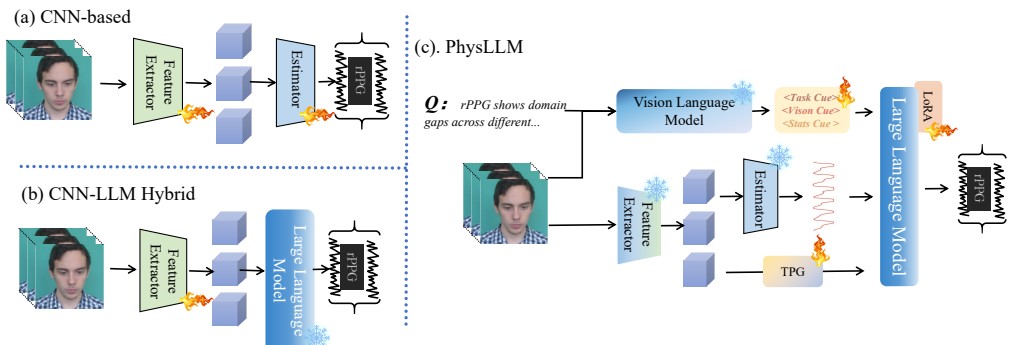

Figure 1: The comparison between (a) typical CNN-based rPPG model, (b) pure LLM model, and (c) our proposed PhysLLM.

Li et al., 2023; Lu et al., 2023; Niu et al., 2019; Špetlík et al., 2018; Chen & McDuff, 2018; Liu et al., 2021b) and Transformers (Yu et al., 2022; Shao et al., 2023; Qian et al., 2024; Yu et al., 2023), have been proposed to address some of these challenges by learning more complex representations of facial features. Fig. 1(a) shows a typical CNN-based framework where features extracted by the encoder are directly fed into the estimator to produce the final sequence. However, these methods are still sensitive to visual noise (e.g., motion blur, occlusions, and low resolution) and often rely on a single video stream, which limits robustness in real-world settings. Introducing textual descriptions provides complementary context about scene conditions—such as occlusion, motion artifacts, and lighting changes—so the model can adapt its processing accordingly. This multi-modal design lets the LLM combine visual cues with semantic context, improving physiological signal extraction under challenging conditions.

Large Language Models (LLMs) offer significant advantages for enhancing rPPG methods, particularly in modeling long-term temporal dependencies (Tang et al., 2025; Jin et al., 2023). While traditional rPPG approaches struggle with extended video sequences, LLMs excel at capturing complex sequential patterns, making them promising candidates for physiological signal estimation. Previous works like TimeLLM (Jin et al., 2023) have demonstrated the effectiveness of LLMs in time-series applications, suggesting untapped potential for rPPG tasks. However, directly applying LLMs to rPPG estimation presents fundamental challenges, as illustrated in Fig. 1(b). The mismatch between LLMs' discrete operations and rPPG features' continuous nature leads to poor representations and high noise sensitivity. Despite LLMs' proven capabilities in cross-modal tasks, their text-oriented architecture requires significant adaptation for effective physiological signal analysis.

To address these limitations, we propose PhysLLM, a collaborative optimization framework that integrates LLMs with specialized rPPG processing components. As shown in Fig. 1(c), to combine CNN's local spatio-temporal feature extraction with LLM's superior long-range temporal reasoning for more robust physiological measurements, we employ a CNN-based rPPG as the base model, leveraging both the multi-scale features and the estimated rPPG signals to guide LLM learning. Specifically, we introduce a Dual-Domain Stationary (DDS) Algorithm to stabilize the base model's rPPG output and a Vision Aggregator module to fuse multi-scale hemodynamic features. To bridge the gap between rPPG features and LLM processing, we propose Text Prototype Guidance (TPG), which aligns sequence and multi-scale features with LLM text prototypes. Furthermore, we introduce task-specific cues, i.e., environmental factors, physiological knowledge, and task description, with learnable word vectors to enhance the LLM's understanding of the rPPG context through cross-modal learning, integrating both visual and textual information. Therefore, PhysLLM adapts to varying conditions such as lighting changes and motion artifacts, significantly improving the accuracy and robustness of rPPG measurement. Overall, our contributions include:

- We propose PhysLLM, the first framework integrating LLMs into rPPG measurement to establish interpretable connections between physiological dynamics and contextual semantics. The architecture's inherent capacity for long-sequence dependency modeling enables superior performance in complex real-life scenarios.

- We propose a novel time-frequency Dual-Domain stationary (DDS) Algorithm to address spectral-temporal instability through adaptive coefficient modulation with exponential decay characteristics. DDS ensures the processed time series to maintain periodic consistency while reducing noise interference.

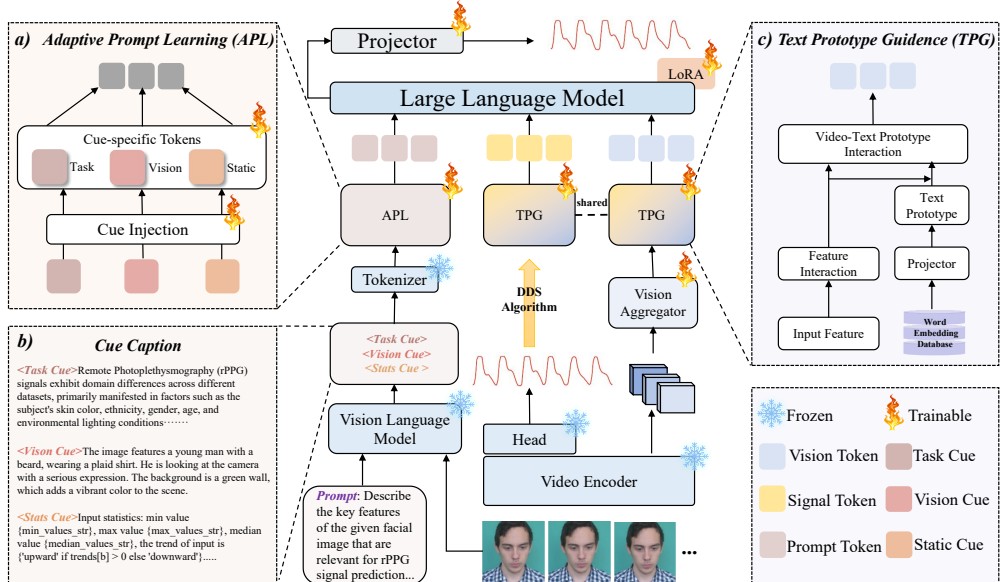

Figure 2: Framework of the PhysLLM. The architecture of PhysLLM comprises three principal data streams that operate in concert. The leftmost stream represents the Physiological Cue-Aware Prompt Learning module, which incorporates task-specific prior knowledge through adaptive prompt learning while generating context-aware prompt tokens. The central and rightmost streams collectively form the Text-Vision-Sequence Embedding Generation pipeline, which integrates our novel Dual-Domain Stationary (DDS) Algorithm and Text Prototype Guidance (TPG) module. This integrated approach facilitates the extraction of sequence tokens from temporal physiological data and visual tokens from facial imagery, both guided by LLM-generated text prototypes that serve as semantic anchors for cross-modal alignment.

- We design task-specific cues to inject physiological priors through physiological statistics, environmental context and task description, enabling dynamic adaptation to challenging scenarios.
- We propose the Text Prototype Guidance (TPG) strategy to establish cross-modal alignment by projecting hemodynamic features into LLM-interpretable semantic space, significantly reducing the gap between sequential, visual, and textual modalities.
- Extensive experiments demonstrate the superiority of PhysLLM even under scenarios with serious degradation.

## 2 RELATED WORK

**Remote Physiological Measurement.** rPPG has advanced significantly from early signal processing methods to deep learning-based approaches. Traditional techniques, such as blind source separation and skin reflection-based models (Poh et al., 2010; Madej et al., 2011; De Haan & Jeanne, 2013; Wang et al., 2017), initially aimed to isolate physiological signals but struggled with real-world variability. Later methods incorporated physiological priors to improve robustness (Zhang et al., 2024), yet they remained sensitive to noise and illumination. With the rise of deep learning, more recent approaches to rPPG measurement have leveraged the power of neural networks to capture rich spatio-temporal representations. Early deep learning-based methods introduced end-to-end spatiotemporal networks (Špetlík et al., 2018; Liu et al., 2020; Yu et al., 2019; Chen & McDuff, 2018; Niu et al., 2019; Liu et al., 2021b; Yu et al., 2023; 2022). These models demonstrated superior performance compared to traditional methods, as they were capable of capturing intricate temporal and spatial relationships within the video data. One major limitation is their sensitivity to visual interference, such as motion blur, occlusions, and varying lighting conditions, all of which can compromise the quality of the extracted rPPG signal. Recent advancements in vision-language multi-modal learning (Yue et al., 2024) have shown great promise in enhancing robustness by aligning with additional modalities for rPPG measurement.

**LLMs for Time Series Tasks.** Recent advancements in LLMs have demonstrated remarkable potential in time series analysis tasks, particularly through their ability to learn temporal patterns and perform cross-modal reasoning. Several pioneering works have explored reprogramming LLMs for time series forecasting without architectural modifications. For instance, Time-LLM (Jin et al., 2023) introduces a novel reprogramming framework that aligns time series embeddings with LLM token spaces using learnable prompt tokens, achieving state-of-the-art performance across multi-

ple domains. This approach builds upon foundational insights from Time Series Forecasting with LLMs (Gruver et al., 2023), which systematically evaluates LLMs' inherent temporal reasoning capabilities and proposes specialized fine-tuning strategies to enhance their forecasting accuracy. In healthcare applications, LLMs have shown particular promise for physiological time series interpretation (Liu et al., 2023b). While existing research primarily focuses on standard physiological signals like ECG and EEG, the application of LLMs to rPPG analysis remains underexplored. Current approaches for rPPG-based health monitoring typically rely on specialized neural architectures, potentially overlooking the cross-modal generalization capabilities inherent in foundation models. Our work bridges this gap by adapting the time series reprogramming module while incorporating rPPG-aware vision-language knowledge, enabling LLMs to interpret subtle cardiovascular patterns from facial videos.

## 3 Methodology

The overall architecture of the proposed PhysLLM is designed to integrate textual, visual, and signal information into a unified framework, enabling robust rPPG physiological estimation. As illustrated in Fig. 2, the framework consists of two key components: Text-Vision-Sequence Embedding Generating (the right part of Fig. 2) and Physiological Cue-Aware Prompt Learning (the left part of Fig. 2). The detailed components of the model and the training objectives are described below.

### 3.1 Text-Vision-Sequence Embedding Generating

While current deep learning frameworks struggle to extract reliable rPPG signals from facial videos due to environmental interference, existing physiological priors in models like PhysNet (Yu et al., 2019) demonstrate inherent robustness to such variations. Building on this foundation, we establish a novel architecture with: 1) A fixed PhysNet (Yu et al., 2019) backbone preserving raw rPPG signals and spatio-temporal features, 2) Three trainable enhancement modules (Dual-Domain Stationary Algorithm module for stable signal tokenization, dedicated multi-scale interaction module for hierarchical feature extraction, Text Prototype Guidance module aligning these representations with LLM semantic spaces) that progressively refine the physiological representations.

**Dual-Domain Stationary (DDS) Algorithm.** To effectively reduce the interference of noise on the model and further enhance its robustness and prediction accuracy, we propose a novel algorithm. This algorithm optimizes the signal processing workflow in a targeted manner, significantly improving the quality of rPPG signals and providing a more reliable data foundation for subsequent analysis.

Specifically, let $x \in \mathbb{R}^{B \times L}$ denote the raw rPPG waveform extracted by the PhysNet backbone (Yu et al., 2019), where batch size $B$ and sequence length $L$ inherit the output dimensions of the backbone. We first compute the global mean $\mu$ and global standard deviation $\sigma$ across the entire sequence,

The normalized output $x'$ is then calculated as:
$$x' = \frac{x - \mu}{\sigma + \epsilon}, \quad \epsilon = 10^{-5}. \tag{1}$$

We denote similar standardization processes in a unified manner as function $\mathcal{H}(\cdot)$. To ensure global stationarity, we proceed temporal smoothing with the following operations:
$$z_i^{time} = \alpha \cdot x_i' + (1 - \alpha) \cdot z_{i-1}, \quad z_0 = x_0', \tag{2}$$
where i denotes the current frame value, and $\alpha$ is a smoothing factor. The proof of stationarity is provided in Appendix. C. In the following text, we denote these smoothing operations collectively as $\mathcal{Z}(\cdot)$. Therefore, the above operation can be simplified as $z^{time} = \mathcal{Z}(x')$.

In parallel, the module performs frequency domain decomposition using the discrete wavelet transform (DWT). The DWT decomposes the input signal into approximation coefficients ($ac$) and detail coefficients ($dc$) on multiple scales. Specifically, the results can be expressed as:
$$x_{ac}, [x_{dc,1}, \ldots, x_{dc,J}] = \text{DWT}(x), \tag{3}$$
where $J$ is the decomposition level ($J = 3$ in this implementation). After normalization, the inverse wavelet transform (IDWT) reconstructs the smoothed frequency-domain representation $z^{fre}$:
$$z^{fre} = \text{IDWT}(\mathcal{Z}(\mathcal{H}(x_{ac})), [\mathcal{Z}(\mathcal{H}(x_{dc,1})), \ldots, \mathcal{Z}(\mathcal{H}(x_{dc,J}))]). \tag{4}$$
To combine the advantages of time-domain and frequency-domain processing, an adaptive weighting mechanism is introduced. The final smoothed output $x_{rec}$ is computed as:
$$z = (1 - \beta) \cdot z^{time} + \beta \cdot z^{fre}, \tag{5}$$
where $\beta \in [0, 1]$ is a learnable parameter.

**Multi-scale Interaction via Vision Aggregator (VA).** We introduce the multi-scale Interaction module to effectively integrate multi-scale features from different modalities. As shown in Fig. 3, this module leverages a combination of cross-attention and self-attention mechanisms to capture both inter-modal and intra-modal relationships between high-level and low-level features. By employing learnable scaling parameters, the module ensures that the fused feature representation is adaptive and context-aware.

Specifically, we extract $M$ layers features from the backbone, denoted as $\mathcal{F} = [f_1, f_2, f_3, ..., f_M]$, in which $f_i \in \mathbb{R}^{B \times T \times H \times W}$. To align these features into a common embedding space, we employ a set of feature projection layers, $F_i = \text{Projection}(f_i, l_{target})$, where $l_{target}$ is the setting length after compression.

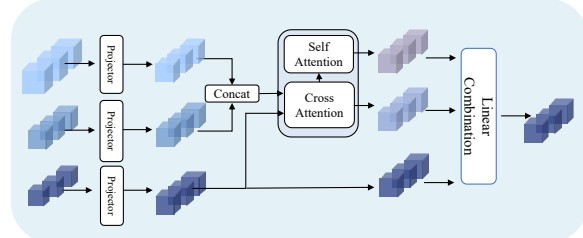

Figure 3: Architecture of the Vision Aggregator. It employs a hierarchical attention architecture to dynamically synthesize multi-scale feature representations.

Since the strong semantic alignment between deep visual features and text space, we use the deep feature $F_M$ as queries to dynamically extract missing details from shallow features $X = \text{Concat}(F_1, F_2, \ldots, F_{M-1})$. This results in a visual feature $F_{cross} \in R^{B \times T \times D}$ with richer fine-grained features. This process can be formulated as:

$$F_{cross} = \text{CrossAttention}(F_M, X, X), \tag{6}$$

where $\text{CrossAttention}(\cdot)$ denotes the cross-attention mechanism, $\text{Concat}(\cdot)$ represents the concatenation operation.

To enhance the representation by capturing internal dependencies within the cross-attended feature, we incorporate a self-attention mechanism into the feature map $F_{self}$, formulated as:

$$F_{self} = \text{SelfAttention}(F_{cross}). \tag{7}$$

To combine the outputs of the cross- and self-attention mechanisms, the final fused features are computed as:

$$F_{visual} = F_M + \gamma_2 \cdot (F_{cross} + \gamma_1 \cdot F_{self}), \tag{8}$$

where $\gamma_1, \gamma_2$ are the learnable vector.

**Text Prototype Guidance (TPG).** To fully leverage the prior knowledge embedded in rPPG signals and visual features, we aim to enable these modalities to play a critical role in the task. However, rPPG signals and visual features are neither directly editable nor easily describable in natural language without missing information, which poses significant challenges for guiding LLMs to understand temporal and visual features without resource-intensive fine-tuning. Specifically, the non-discrete and high-dimensional nature of such data makes it difficult to transform them into symbolic representations suitable for language models.

As shown in Fig. 2(c), to bridge this gap, we propose to reprogram the rPPG signals and visual features with word embeddings $E \in \mathbb{R}^{V \times D}$, where $V$ is the vocabulary size. Nevertheless, there is no prior knowledge indicating which source tokens are directly relevant. Thus, simply leveraging $E$ will resulting large and potentially dense reprogramming space. A simple solution is to maintain a small collection of text prototypes by linearly probing, denoted as $E' \in \mathbb{R}^{V' \times D}$, where $V' \ll V$.

The text prototypes $E'$ are then used to enhance the interaction between rPPG signals, visual features, and language model. To extract more sufficient details from the input video and signal, we devise a block, which consists of multiple different transformer layers. Specifically, given input feature $X$:

$$X_{self} = \text{SelfAttention}(X), \tag{9}$$

$$E'_{fusion} = E' + X_{self}, \tag{10}$$

$$y_2(E'_{fusion}; X) = \text{CrossAttention}(E'_{fusion}, X, X), \tag{11}$$

$$Output = E'_{fusion} + y_2(E'_{fusion}; X), \tag{12}$$

$$\mathcal{T}_{out} = FFN(E'_2), \tag{13}$$

where $E'_1$ and $E'_2$ are the updated versions of $E'$. It is worth noting that the TPG module guides not only visual features, but also sequence features, so the input feature X here has the two meanings. In

addition, the visual feature part and the sequence feature part share the same TPG module to learn potential associations.

For simplicity, we represent this sequence of operations as a single function $TPG(\cdot)$, so the output of this module can be expressed as $\mathcal{T}_{out} = TPG(X)$.

### 3.2 Physiological Cue-Aware Prompt Learning

The extraction of reliable rPPG signals from facial video data presents significant challenges due to varying lighting, subject mobility, and diverse skin tones in complex real-world scenarios. To address these challenges through cross-modal learning, we propose a Physiological Cue-Aware Prompt Learning framework that generates rPPG sample-specific cues for guiding adaptive token learning.

**Cue Caption.** Prompt-based adaptation enables LLMs to address specialized subtasks without requiring parameter updates, as demonstrated by the zero-shot capabilities of architectures like LLaMA (Touvron et al., 2023) and DeepSeek (Guo et al., 2025). To eliminate manual annotation of context-aware descriptors, we leverage LLaVA (Liu et al., 2023a) for automated physiological cue generation through structured prompt engineering, as shown in Fig. 2(b). For rPPG-specific adaptation, our prompts focus on three critical visual domains: 1) facial anatomical characteristics, 2) transient emotional expressions, and 3) environmental illumination dynamics, as exemplified by our designed query templates in Fig. 4.

Formally, given an input video sequence $V \in \mathbb{R}^{3 \times T \times H \times W}$, we extract the central frame $I_t$ and process it through LLaVA with task-oriented prompts to obtain detail description, and tokenize the description, $C_{vision} = Tokenizer(LLaVA(I_t, Q))$, where $Q$ denotes our physiological cue queries.

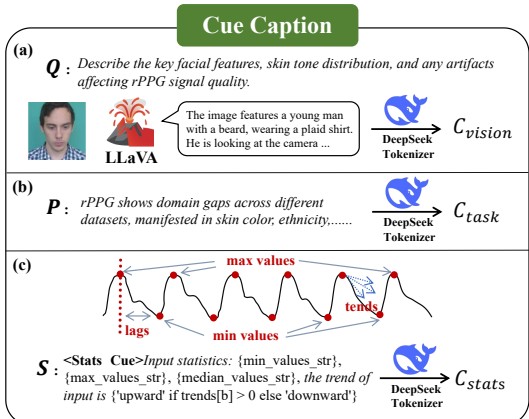

Figure 4: Introduction to the composition of cues. (a) Extracting visual priors (e.g., lighting, facial expressions, occlusions) via LLaVA and encoding them into visual tokens. (b) Tokenizing textual descriptions of the rPPG task to derive task-specific priors. (c) Analyzing statistical features of rPPG signals from the backbone network to generate statistical prior tokens. These cues synthesize visual, semantic, and statistical priors for enhanced physiological signal analysis.

Furthermore, we formalize domain knowledge through task-specific primers derived from consensus descriptions in rPPG literature. These primers are encoded into LLM-compatible tokens via the model's native tokenizer:

$$C_{task} = Tokenizer(\mathcal{P}), \tag{14}$$

where $\mathcal{P}$ represents the standardized task preamble.

While traditional text-based prompting strategies have demonstrated effectiveness in general vision-language tasks, we identify critical limitations when applied to physiological signal estimation: (1) Static visual descriptors cannot sufficiently capture temporal hemodynamic variations, and (2) Task-agnostic queries fail to address rPPG-specific contextual dependencies. To bridge this semantic gap, we propose statistical priors derived from pretrained rPPG models as physiological knowledge supplements. Although ground truth signals remain unavailable during inference, the pretrained network's output sequences (discussed in Section 3.1) preserve valid distribution characteristics. So the statistical cue $\mathcal{S}$ is derived from the pretrained model's rPPG signal projections $x_{enc} \in \mathbb{R}^{B \times T}$, where $B$ denotes batch size and $T$ the temporal dimension. For each sample $b \in [1, B]$, we compute,

$$\mathcal{S} = \left\{ \min_t(x_{enc}[b,t]), \ \max_t(x_{enc}[b,t]), \ \text{median}(x_{enc}[b,:]), \right.$$
$$\left. \Gamma(x_{enc}[b,:]), \ \text{sgn}\left(\sum_{t=1}^{T-1}(x_{enc}[b,t+1] - x_{enc}[b,t])\right), \text{TopK}\left(\mathcal{L}(x_{enc}[b,:]),5\right) \right\} \tag{15}$$

where $\Gamma(\cdot)$ calculates signal trends through first-order differencing, $\Gamma(\mathbf{x}) = \sum_{i=2}^{T}(\mathbf{x}_i - \mathbf{x}_{i-1})$.

Then $\mathcal{S}$ will make up the symbolic representation as described in Fig. 4(c). Finally, the tokenization process is formulated as $C_{stats} = \text{Tokenizer}(\mathcal{S})$.

Table 1: Intra-dataset testing results on UBFC-RPPG (Bobbia et al., 2019), PURE (Stricker et al., 2014), BUAA (Xi et al., 2020) and MMPD (Tang et al., 2023) datasets. Best results are in **bold**.

| Method | UBFC-rPPG | | | PURE | | | BUAA | | | MMPD | | |
|---|---|---|---|---|---|---|---|---|---|---|---|---|
| | MAE↓ | RMSE↓ | R↑ | MAE↓ | RMSE↓ | R↑ | MAE↓ | RMSE↓ | R↑ | MAE↓ | RMSE↓ | R↑ |
| ICA (Poh et al., 2010) | 16.00 | 25.65 | 0.44 | 4.77 | 16.07 | 0.72 | - | - | - | 18.60 | 24.30 | 0.01 |
| CHROM (de Haan & Jeanne, 2013) | 4.06 | 8.83 | 0.89 | 5.77 | 14.93 | 0.81 | - | - | - | 13.66 | 18.76 | 0.08 |
| Green (Verkruysse et al., 2008) | 19.73 | 31.00 | 0.37 | 10.09 | 23.85 | 0.34 | 6.89 | 10.39 | 0.60 | 21.68 | 27.69 | -0.01 |
| POS (Wang et al., 2017) | 4.08 | 7.72 | 0.92 | 3.67 | 11.82 | 0.88 | - | - | - | 12.36 | 17.71 | 0.18 |
| Meta-rPPG (Lee et al., 2020) | 5.97 | 7.42 | 0.57 | 2.52 | 4.63 | 0.98 | - | - | - | - | - | - |
| PhysNet (Yu et al., 2019) | 2.95 | 3.67 | 0.97 | 2.10 | 2.60 | 0.99 | 10.89 | 11.70 | -0.04 | 4.80 | 11.80 | 0.60 |
| PhysFormer (Yu et al., 2022) | 0.92 | 2.46 | 0.99 | 1.10 | 1.75 | 0.99 | 8.46 | 10.17 | -0.06 | 11.99 | 18.41 | 0.18 |
| EfficientPhys (Liu et al., 2021a) | 1.41 | 1.81 | 0.99 | 4.75 | 9.39 | 0.99 | 16.09 | 16.80 | 0.14 | 13.47 | 21.32 | 0.21 |
| Contrast-Phys+ (Sun & Li, 2024) | 0.21 | 0.80 | 0.99 | 0.48 | 0.98 | 0.99 | - | - | - | - | - | - |
| RhythmFormer Zou et al. (2025) | 0.50 | 0.78 | 0.99 | 0.27 | 0.47 | 0.99 | 9.19 | 11.93 | -0.10 | 4.69 | 11.31 | 0.60 |
| **PhysLLM (Ours)** | **0.21** | **0.57** | **0.99** | **0.17** | **0.35** | **0.99** | **6.48** | **8.48** | **0.63** | **4.36** | **10.76** | **0.65** |

**Adaptive Prompt Learning (APL).** Conventional multimodal fusion approaches typically employ static weighting coefficients for prompt integration, suffering from two limitations: (1) fixed combination ratios cannot adapt to learnable LLM, and (2) inability to adaptively select the key cues across different data. As shown in Fig. 2(a), our adaptive fusion paradigm addresses these constraints through learnable hierarchical composition. Formally, having obtained three fundamental cue captions $C = \{C_{task}, C_{vision}, C_{stats}\}$, we first implement modality-specific conditioning via dedicated transformation networks,

$$\mathcal{E}_k = \text{AttentiveCompressor}_k(C_k), \quad k \in \{\text{ task, vision, stats }\}, \tag{16}$$

where each compressor employs temporal-contextual attention to maintain modality-specific patterns,

$$\text{AttentiveCompressor}_k(x) = \text{Softmax}\left(\frac{Q_k K_k^T}{\sqrt{d}}\right) V_k, \tag{17}$$

with $Q_k, K_k, V_k$ derived from $x$ through linear projections.

In addition, the adaptive fusion mechanism learns three independent parameter matrices:

$$\mathbf{W} = [W_{\text{task}}, W_{\text{vision}}, W_{\text{stats}}] \in \mathbb{R}^{3 \times B \times L \times d}, \tag{18}$$

where $L$ is predefined hyperparameter about target prompt sequence length, $d$ is the token embedding dimension, and $W_{\text{task}}, W_{\text{vision}}, W_{\text{stats}}$ denote learnable tokens, respectively.

The complete fusion operation is formalized as:

$$\mathcal{T}_{\text{cue}} = \sum_{k \in \Omega} \mathbf{W}^{(k)} \odot \mathcal{E}_k, \Omega \triangleq \{task, vision, stats\}. \tag{19}$$

### 3.3 TRAINING OBJECTIVES

In Section 3.1, we obtain the stationary signal $z$ and the fused multi-scale visual features $F_{visual}$ through the DDS module and multi-scale interaction, respectively. Through TPG, we can obtain the text-guided vision token $\mathcal{T}_{\text{vision}} = TPG(F_{visual})$, and the text-guided signal token $\mathcal{T}_{\text{signal}} = TPG(z)$. Then, we input $\mathcal{T}_{\text{cue}}, \mathcal{T}_{\text{vision}}, \mathcal{T}_{\text{signal}}$ into LLM and predict waveforms $\hat{y}$. Finally, we employ the mean square error loss function as the total loss,

$$\mathcal{L}_{MSE} = \frac{1}{n} \sum_{i=1}^{n} (y_i - \hat{y}_i)^2, \tag{20}$$

where $y$ is the ground truth PPG signals.

## 4 EXPERIMENTS

### 4.1 DATASETS AND PERFORMANCE METRICS

We comprehensively evaluate models on four benchmark datasets (UBFC-rPPG (Bobbia et al., 2019), PURE (Stricker et al., 2014), BUAA (Xi et al., 2020), and MMPD (Tang et al., 2023)). Following (Sun & Li, 2022; Yu et al., 2019), we calculate mean absolute error (MAE), root mean square error (RMSE), and Pearson's correlation coefficient (R) between the predicted HRs versus the ground-truth HRs as evaluation metrics. Notably, for MAE and RMSE, lower values indicate reduced error margins, whereas for R, values approaching 1.0 signify diminished error. Among them, both MAE and RMSE are measured in terms of bpm (beats per minute). More implementation details can be found in Appendix. B.

Table 2: Cross-domain generalization evaluation with dual-source training and single-target testing.

| Method | P+B → M | | P+U → M | | B+U → M | |
|---|---|---|---|---|---|---|
| | MAE↓ | RMSE↓ | MAE↓ | RMSE↓ | MAE↓ | RMSE↓ |
| Green (Verkruysse et al., 2008) | 21.7 | 27.7 | 21.7 | 27.7 | 21.7 | 27.7 |
| EfficientPhys (Liu et al., 2021a) | 11.9 | 18.5 | 11.8 | 18.9 | 15.5 | 20.8 |
| PhysFormer (Yu et al., 2022) | 13.9 | 18.6 | 11.4 | 17.5 | 13.2 | 16.5 |
| PhysNet (Yu et al., 2019) | 13.2 | 16.7 | 11.0 | 17.3 | 13.5 | 17.0 |
| RhythmFormer (Zou et al., 2025) | 13.98 | 19.5 | 10.5 | 16.7 | 12.6 | 17.5 |
| **PhysLLM (Ours)** | **11.9** | **15.3** | **9.95** | **14.96** | **12.1** | **15.2** |

Table 3: Cross-domain generalization evaluation with three-source training and single-target testing.

| Method | Others → MMPD | | Others → BUAA | |
|---|---|---|---|---|
| | MAE↓ | RMSE↓ | MAE↓ | RMSE↓ |
| Green (Verkruysse et al., 2008) | 21.7 | 27.7 | 6.9 | 10.4 |
| EfficientPhys (Liu et al., 2021a) | 13.2 | 20.02 | 32.3 | 34.0 |
| PhysFormer (Yu et al., 2022) | 13.9 | 19.3 | 7.7 | 12.4 |
| PhysNet (Yu et al., 2019) | 12.8 | 16.3 | 12.8 | 16.4 |
| RhythmFormer (Zou et al., 2025) | 16.1 | 20.5 | 6.04 | 10.8 |
| **PhysLLM (Ours)** | **12.2** | **15.5** | **6.01** | **8.6** |

## 4.2 Intra-dataset Testing

We first evaluate the HR estimation on all datasets under intra-dataset setting. We compare our method with 10 methods, including traditional methods and deep learning methods. Table 1 displays intra-dataset testing results for UBFC-rPPG (Bobbia et al., 2019), PURE (Stricker et al., 2014), BUAA (Xi et al., 2020) and MMPD (Tang et al., 2023).

**HR Estimation on UBFC-rPPG (Bobbia et al., 2019).** On UBFC-rPPG (Bobbia et al., 2019), we follow (Luo et al., 2024) by training on the first 30 subjects and testing on the remaining 12. As shown in Table 1, PhysLLM achieves state-of-the-art HR estimation with MAE 0.21 bpm, RMSE 0.57 bpm, and R 0.99. These results indicate stable rPPG fitting and effective long-term HR tracking.

**HR Estimation on PURE (Stricker et al., 2014).** Following (Luo et al., 2024), we compare PhysLLM with 9 methods. As shown in Table 1, PhysLLM achieves the best HR performance across all metrics, outperforming the second-best PhysFormer (Yu et al., 2022) by 0.31 bpm MAE and 0.63 bpm RMSE, demonstrating strong robustness to head-motion interference.

**HR Estimation on BUAA (Xi et al., 2020).** It is partitioned in sequence into training and test sets with a ratio of 7:3. The performance of existing methods on this dataset is reimplemented by ourselves with the rPPG toolbox (Liu et al., 2023c). The HR estimation results are shown in Table 1. The proposed PhysLLM outperforms the existing state-of-the-art methods on MAE (6.48 bpm), RMSE (8.48 bpm), and R (0.63) metrics for HR prediction. The results show the strong robustness of PhysLLM against various lighting condition disturbances.

**HR Estimation on MMPD (Tang et al., 2023).** Following the protocol in (Zou et al., 2024), we compared our method against existing state-of-the-art approaches, as depicted in Table 1. The proposed PhysLLM outperforms the existing state-of-the-art methods on MAE (4.36 bpm), RMSE (10.76 bpm), and R (0.65) metrics for HR prediction. These results demonstrate that PhysLLM performs well under real-world conditions.

## 4.3 Cross-dataset Testing

We also perform cross-dataset testing to assess the generalization capability of PhysLLM. We conduct both Two-training and One-testing protocol and Three-training and One-testing protocol. In each case, we train on relatively simple datasets and evaluate performance on one more complex dataset.

**Dual-source training and single-target testing protocol.** To evaluate generalization, we train on two datasets and test on the most challenging MMPD (Tang et al., 2023). For instance, $P + U \rightarrow M$ denotes training on PURE (Stricker et al., 2014) and BUAA (Xi et al., 2020) and testing on MMPD using the full datasets. As shown in Table 2, PhysLLM remains best under these cross-domain settings, reducing MAE/RMSE by 1.05/2.34 bpm on $P + U \rightarrow M$, demonstrating strong generalization enabled by visual context and LLM modeling.

**Three-source training and single-target testing protocol.** To further evaluate whether PhysLLM can learn domain invariance knowledge without mixing domain-specific knowledge, we trained on three datasets and tested on the remaining one dataset. For example, $Others \rightarrow MMPD$ means training on the other datasets (UBFC-rPPG (Bobbia et al., 2019), PURE (Stricker et al., 2014), BUAA (Xi et al., 2020)) and test on MMPD (Tang et al., 2023). It is worth noting that we only tested two difficult datasets: BUAA (Xi et al., 2020) and MMPD (Tang et al., 2023). It can be seen from the results in Table 3 that PhysLLM achieves

Table 4: Ablation results of the main components on the UBFC-rPPG (Bobbia et al., 2019) dataset.

| Method | | | UBFC-rPPG | | |
|---|---|---|---|---|---|
| DDS | VA | TPG | MAE↓ | RMSE↓ | R↑ |
| ✓ | × | × | 0.36 | 1.12 | 0.98 |
| × | ✓ | × | 0.41 | 1.26 | 0.98 |
| × | × | ✓ | 0.32 | 1.00 | 0.98 |
| ✓ | ✓ | × | 0.27 | 0.92 | 0.98 |
| ✓ | × | ✓ | 0.34 | 1.05 | 0.99 |
| × | ✓ | ✓ | 0.25 | 0.76 | 0.99 |
| ✓ | ✓ | ✓ | **0.21** | **0.57** | **0.99** |

Table 5: Ablation experiments of different large language models and transformer baseline. While DeepSeek is used as the default LLM in our experiments, the proposed approach can be easily generalized to other large language models.

| Method | LLM Param. | UBFC | | PURE | |
|---|---|---|---|---|---|
| | | MAE↓ | RMSE↓ | MAE↓ | RMSE↓ |
| PhysNet (Yu et al., 2019) | - | 2.95 | 3.67 | 2.10 | 2.60 |
| PhysLLM (w. Sundial (Liu et al., 2025)) | - | 0.92 | 2.46 | 3.22 | 7.54 |
| PhysLLM (w. DeepSeek (Guo et al., 2025)) | 1.5B | 0.21 | 0.57 | 0.17 | 0.35 |
| PhysLLM (w. Bert (Devlin et al., 2019)) | 0.11B | 0.19 | 0.76 | 0.43 | 0.80 |
| PhysLLM (w. GPT2 (Lagler et al., 2013)) | 0.124B | 0.19 | 0.76 | 0.14 | 0.35 |

the best performance in both datasets, and even outperforms the existing state-of-the-art methods on MAE (6.0 bpm), RMSE (8.6 bpm), and R (0.63) metrics for HR prediction on $Others \rightarrow BUAA$. This shows that PhysLLM will not confuse domain-specific knowledge even if trained on three datasets, and can stably learn domain-invariant rPPG knowledge.

## 4.4 ABLATION STUDY

**Impact of Dual-Domain Stationary Algorithm (DDS).** The role of DDS is to transform the input signal $x$ into a stationary signal $Z(x)$. To validate the effectiveness of the DDS, we construct a comparative experiment designed to remove DDS. Specifically, we directly use the rPPG signal from the video encoder, instead of the rPPG signal smoothed by DDS. The results are shown in the sixth row of Table 4. DDS reduces the MAE by 0.04 bpm, and the RMSE by 0.19 bpm on PURE dataset.

**Impact of Vision Aggregator (VA).** In this ablation study, we simply remove the Vision Aggregator to evaluate its impact on the model. As shown in the results on UBFC-rPPG dataset, the absence of the Vision Aggregator leads to a noticeable decrease in performance, underscoring its importance in the model architecture. The comparison between the fifth and seventh rows in Table 4 clearly demonstrates that the fusion of visual features is effective.

**Impact of Text Prototype Guidance (TPG).** To evaluate the effectiveness of text prototype guidance, we conducted an ablation study by removing the text prototype guidance. The results on UBFC-rPPG dataset are shown in the fourth row of Table 4. TPG brings a significant decrease in MAE (0.06 bpm) and RMSE (0.35 bpm). These results underscore the importance of TPG in facilitating the fusion of quasi-periodic video features.

**Impact of prompt components.** To evaluate the contributions of different prompt components (Vision, Statics, Task, and adaptive learning) in PhysLLM's cue caption and adaptive prompt learning, we conduct an ablation study and report MAE/RMSE in Table 6. The full configuration achieves the best performance (MAE 0.21 bpm, RMSE 0.57 bpm), yielding a 76% improvement over the worst setting (MAE 1.31 bpm, RMSE 2.10 bpm) that removes adaptive learning, confirming its critical role. We also observe that adding the Task component

Table 6: Ablation study of prompt components on the UBFC-rPPG (Bobbia et al., 2019) dataset.

| Method | | | | UBFC-rPPG | |
|---|---|---|---|---|---|
| Vision | Stats | Task | Adaptive learning | MAE↓ | RMSE↓ |
| ✓ | ✗ | ✗ | ✓ | 0.83 | 1.53 |
| ✗ | ✓ | ✗ | ✓ | 0.71 | 2.26 |
| ✗ | ✗ | ✓ | ✓ | 0.84 | 2.41 |
| ✓ | ✓ | ✗ | ✓ | 0.73 | 2.29 |
| ✓ | ✗ | ✓ | ✓ | 0.43 | 1.20 |
| ✗ | ✓ | ✓ | ✓ | 0.61 | 1.53 |
| ✓ | ✓ | ✓ | ✗ | 1.31 | 2.10 |
| ✓ | ✓ | ✓ | ✓ | **0.21** | **0.57** |

generally improves performance, and Vision+Task outperforms Statics+Task, suggesting visual prompts better align with rPPG-specific cues.

**Ablation Study on the LLM Component.** To validate the necessity of using a pre-trained LLM, we compare three LLMs with varying sizes: DeepSeek (Guo et al., 2025), Bert (Devlin et al., 2019), and GPT2 (Lagler et al., 2013). As shown in Table 5, all LLMs achieve competitive performance, with DeepSeek performing best (MAE: 0.21/0.17 on UBFC/PURE). We further replace the LLM with Sundial (Liu et al., 2025), a temporal transformer without LLM pre-training. The significant performance drop (MAE: 3.35/4.05) demonstrates that the LLM's pre-trained knowledge is essential for cross-dataset generalization, not merely the transformer architecture.

## 4.5 VISUALIZATION AND DISCUSSION

**Robustness Analysis on Skin Tones and Lighting Conditions.** To evaluate robustness, we perform stress tests on the MMPD (Tang et al., 2023) dataset by varying skin tones (Types 3–6) and lighting

Table 7: Performance Comparison across Different Skin Tones and Lighting Conditions on MMPD (Tang et al., 2023) dataset (MAE/RMSE).

(a) Performance across Different Skin Tones.

| Method | Type 3 | Type 4 | Type 5 | Type 6 |
|---|---|---|---|---|
| PhysFormer (Yu et al., 2022) | 6.11/11.21 | 5.92/10.00 | 5.87/13.12 | 6.81/11.21 |
| RhythmFormer (Zou et al., 2025) | 5.12/11.23 | 5.46/9.14 | 6.32/11.87 | 6.26/9.99 |
| **PhysLLM (Ours)** | **4.96/10.27** | **4.73/8.49** | **4.99/11.07** | **5.73/8.49** |

(b) Performance across Different Lighting Conditions.

| Method | LED-Low | LED-High | Incandescent | Natural |
|---|---|---|---|---|
| PhysFormer (Yu et al., 2022) | 6.36/11.72 | 5.12/9.39 | 5.71/12.73 | 6.61/11.36 |
| RhythmFormer (Zou et al., 2025) | 5.85/11.71 | 4.46/8.98 | 3.64/11.87 | 5.65/12.31 |
| **PhysLLM (Ours)** | **4.46/9.57** | **3.72/8.10** | **3.59/11.43** | **3.45/6.81** |

Table 8: Comparison of Model Complexity

| Method | Params (M) | MACs (G) |
|---|---|---|
| TS-CAN (Liu et al., 2020) | 7.5 | 96.0 |
| PhysNet (Yu et al., 2019) | 0.77 | 56.1 |
| DeepPhys (Chen & McDuff, 2018) | 7.5 | 96.0 |
| EfficientPhys (Liu et al., 2021a) | 7.4 | 45.6 |
| PhysFormer (Yu et al., 2022) | 7.38 | 40.5 |
| RhythmFormer (Zou et al., 2025) | 4.21 | 28.8 |
| Contrast-phys+ (Sun & Li, 2024) | 0.85 | 145.7 |
| PhysMamba (Luo et al., 2024) | 0.56 | 47.3 |
| **PhysLLM (Ours)** | **97.2** | **424.3** |

conditions (LED-Low/High, Incandescent, Natural). As shown in Table 7, PhysLLM consistently outperforms PhysFormer and RhythmFormer across all settings, especially under extreme lighting (MAE/RMSE: 3.45/6.81 in natural light) and diverse skin tones (4.73/8.49 for Type 4). Overall, PhysLLM remains stable under challenging variations, supporting its real-world applicability across different populations and environments.

**Model Complexity Analysis.** As shown in Table 8, PhysLLM has higher computational complexity (97.2M parameters, 424.3G MACs) due to the LLM backbone. While we acknowledge this overhead, it is a necessary trade-off for achieving superior cross-dataset generalization and robustness demonstrated in our experiments. Future work will explore model compression techniques such as knowledge distillation and parameter-efficient fine-tuning to reduce costs while maintaining performance for resource-constrained deployment.

**Visualization of saliency maps.** Following (Sun & Li, 2022), the saliency maps of the PhysLLM are visualized in Fig. 5 using samples from UBFC-rPPG (Bobbia et al., 2019) and PURE (Stricker et al., 2014) datasets. The more obvious the red-green area, the more attention it means. We selected images with certain features from the dataset to test rPPG tasks, such as head rotation, with a noticeable beard, darker skin tone, and wearing glasses.

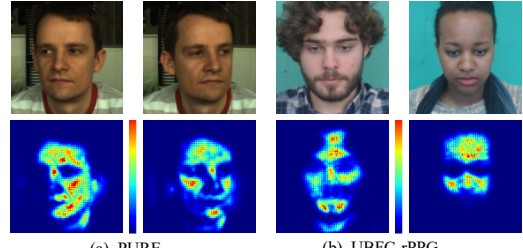

(a) PURE    (b) UBFC-rPPG

Figure 5: Visualization of saliency maps from PhysLLM on PURE and UBFC-rPPG datasets.

From the test results, it can be seen that PhysLLM can effectively capture rPPG-related parts. For example, the observable parts are concentrated on the cheeks and forehead, which is consistent with the relevant prior knowledge of rPPG. At the same time, relevant parts can effectively avoid hair and obstruction. For example, there is no prominent attention in the area where the hair appears, and even subtle changes in the neck can be observed. This is sufficient to demonstrate the robustness and effectiveness of PhysLLM in various scenarios.

## 5  CONCLUSION

In this paper, we propose the PhysLLM, a framework that integrates LLMs with specialized rPPG components. The Text Prototype Guidance strategy bridges the cross-modal gap, while the Dual-Domain Stationary Algorithm addresses signal instability. Task-specific priors enhance adaptability in challenging scenarios. Experimental results across four datasets show PhysLLM achieves superior accuracy and robustness, particularly under variable illumination and motion. Future work includes improving cross-modal alignment and developing lightweight models for edge deployment.

## ACKNOWLEDGMENTS

This work was supported by the National Natural Science Foundation of China (Grant No. 62306061, 62576076, and 62276170), the CCF-Tencent Rhino-Bird Open Research Fund, the Basic and Applied Basic Research Foundation of Guangdong Province (No. 2021B1515140031), and the Dongguan Social Development Science and Technology Project (Grant No. 20221800906352, 20221800903641, 20211800903402, and 20231800936072). This work was also supported by the Science and Technology Development Fund of Macao (No. 0009/2025/ITP1) and the Macao Polytechnic University Grant (RP/FCA-16/2025). Additionally, this work was also supported by the Intelligent Computing Center of Shenzhen University.

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

# A    INTRODUCTION TO THE DATASETS

**UBFC-rPPG (Bobbia et al., 2019)** contains 42 RGB facial videos from 42 distinct subjects. Each video is captured at 640×480 pixel resolution and 30 frames per second (fps). Recordings take place under varied lighting conditions, including natural sunlight and indoor artificial illumination. Ground-truth physiological signals are recorded via a CMS50E pulse oximeter at 60 Hz, ensuring precise temporal alignment for evaluation.

**PURE (Stricker et al., 2014)** comprises 60 high-quality RGB videos collected from 10 subjects performing six different head movement scenarios (stats, talking, translation movements, etc.). Videos are recorded at 30 fps under consistent indoor lighting and controlled background settings, minimizing external interference. Synchronized physiological measurements are obtained using a CMS50E oximeter sampling at 60 Hz. PURE is particularly valuable for evaluating rPPG performance during facial movements.

**BUAA (Xi et al., 2020)** is designed to assess algorithmic robustness across varying illumination intensities. The dataset features video sequences recorded under a range of controlled lighting conditions, from low-light (below 10 lux) to normal brightness. In our experiments, we only utilize videos captured under illumination levels ≥10 lux, as extremely dim lighting introduces significant image degradation requiring specialized enhancement techniques beyond this study's scope.

**MMPD (Tang et al., 2023)** comprises 660 videos, each lasting one minute, collected from 33 subjects with diverse skin tones and gender distributions. Each video is recorded at 30 fps with a resolution of 320×240 pixels, under four distinct lighting conditions (bright, warm, dim, and colored lighting). Subjects perform various daily activities, introducing intra-subject variability and further increasing dataset complexity.

# B    IMPLEMENTATION DETAILS

We conduct experiments on Pytorch and mainly based on the open-source toolkit rPPG-Toolbox (Liu et al., 2023c). For data pre-processing, we crop the face region in the first frame for each video clip and fix the region box in the following frames. Subsequently, we randomly sample a video chunk of 128 frames and resize them into $128 \times 128$ pixels. We use the default hyperparameters settings $\alpha = 0.8$ and $l_{target} = 32$. The backbone of our PhysLLM is based on the pre-trained PhysNet (Yu et al., 2019), where the training strategies follow the methodology outlined in the original paper. In the intra-dataset testing and ablation study, we use the same training data for PhysNet as for PhysLLM, ensuring consistency in the data used. For cross-domain experiments, we use the pre-trained PhysNet model trained on the PURE dataset (Stricker et al., 2014), ensuring no test set overlap and consistent pre-training conditions. Furthermore, we use the DeepSeek-1.5B (Guo et al., 2025) version as the LLM. The PhysLLM is trained with Adam optimizer and the initial learning rate and weight decay are 1e-4 and 5e-5, respectively. We train our model for 20 epochs on a NVIDIA A100 GPU with batch size of 4.

# C    MATHEMATICAL PROOF OF STATIONARITY

To prove that the outputs of the BDCS algorithm are stationary, we first verify the stationarity of the core **Stationary Algorithm**, then extend this proof to the time-domain and frequency-domain outputs, and finally conclude the global stationarity of the combined output.

## STATIONARY ALGORITHM STATIONARITY

We need to verify the three conditions for stationarity in the output of the Stationary Algorithm $Z(x)$:

$$\begin{aligned}
z_i &= \alpha \cdot x_i + (1 - \alpha) \cdot z_{i-1} \\
&= \alpha \cdot x_i + (1 - \alpha) \cdot (\alpha \cdot x_{i-1} + (1 - \alpha) \cdot z_{i-2}) \\
&= \alpha \cdot x_i + \alpha \cdot (1 - \alpha) \cdot x_{i-1} + (1 - \alpha)^2 \cdot z_{i-2} \\
&= \sum_{k=0}^{\infty} \alpha (1 - \alpha)^k \cdot x_{i-k}
\end{aligned} \tag{21}$$

---

**Algorithm 1** DDS

---

**Input:** Time series signal $x \in \mathbb{R}^{b \times n}$, wavelet basis $\psi$, decomposition level $J$.
**Output:** Stabilized signal $z \in \mathbb{R}^{b \times n}$
  **Define** $\mathcal{S}(x_i)$ :
    $\mu \leftarrow \frac{1}{L} \sum_{i=1}^{L} x_i$
    $\sigma \leftarrow \sqrt{\frac{1}{L} \sum_{i=1}^{L} (x_i - \mu)^2}$
    **return** $\frac{x_i - \mu}{\sigma + \varepsilon}$
  // **Time Domain Decomposition**
  **Define** $\mathcal{Z}(x_i') = \alpha \cdot x_i' + (1 - \alpha) \cdot z_{i-1}$
  $x_i' \leftarrow \mathcal{S}(x_i)$ // Standardization
  $z_0^{time} \leftarrow x_0'$
  **for** $i = 1$ to $n$ **do**
    $z_i^{time} \leftarrow \mathcal{Z}(x_i')$
  **end for**
  // **Frequency Domain Decomposition**
  $x_{ac}, [x_{dc,1}, \ldots, x_{dc,J}] \leftarrow \text{DWT}(x, \psi, J)$ // Wavelet decomposition
  $x_{ac}' \leftarrow \mathcal{S}(x_{ac}), x_{dc,j}' \leftarrow \mathcal{S}(x_{dc,j})$ for $j = 1, \ldots, J$
  $z_{ac} \leftarrow \mathcal{Z}(x_{ac}'), z_{dc,j} \leftarrow \mathcal{Z}(x_{dc,j}')$ for $j = 1, \ldots, J$
  $z^{fre} \leftarrow \text{IDWT}(z_{ac}, [z_{dc,1}, \ldots, z_{dc,J}], \psi)$ // Inverse wavelet transform
  // **Adaptive Fusion**
  $z \leftarrow (1 - \beta) \cdot z^{time} + \beta \cdot z^{fre}$
  **return** $z$

---

where $x_i$ is a series with a mean of 0 and a variance of 1. If i ≤ k , $x_{i-k} = 0$

1. CONSTANT MEAN

The normalized signal $x_i$ has zero mean:

$$\mathbb{E}[x_i] = 0. \tag{22}$$

The smoothed signal $z_i$ is defined as:

$$z_i = \sum_{k=0}^{\infty} \alpha(1-\alpha)^k \cdot x_{i-k}. \tag{23}$$

Since $\mathbb{E}[y(t)] = 0$, the mean of $z_i$ is:

$$\mathbb{E}[z_i] = \sum_{k=0}^{\infty} \alpha(1-\alpha)^k \cdot \mathbb{E}[x_{i-k}] = 0. \tag{24}$$

Thus, the mean of $z_i$ is constant and equal to zero.

2. CONSTANT VARIANCE

The variance of $z_i$ is given by:
$$\text{Var}(z_i) = \mathbb{E}[z_i^2] - (\mathbb{E}[z_i])^2. \tag{25}$$

Since $\mathbb{E}[z_i] = 0$, we have:
$$\text{Var}(z_i) = \mathbb{E}[z_i^2]. \tag{26}$$

$$\text{Var}(z_i) = \mathbb{E}\left[\left(\sum_{k=0}^{\infty} \alpha(1-\alpha)^k \cdot x_{i-k}\right)^2\right]. \tag{27}$$

Assuming $x_{i-k}$ has unit variance ($\text{Var}(x_{i-k}) = 1$) and different time points of $x_{i-k}$ are uncorrelated:

$$\text{Var}(z_i) = \sum_{k=0}^{\infty} \left( \alpha(1-\alpha)^k \right)^2 \cdot \text{Var}(x_{i-k}). \tag{28}$$

Since $\text{Var}(y(t)) = 1$, we have:

$$\text{Var}(z_i) = \sum_{k=0}^{\infty} \left( \alpha(1-\alpha)^k \right)^2. \tag{29}$$

This is a geometric series with sum:

$$\begin{aligned}
\sum_{k=0}^{\infty} \left( \alpha(1-\alpha)^k \right)^2 &= \alpha^2 \sum_{k=0}^{\infty} (1-\alpha)^{2k} \\
&= \frac{\alpha^2}{1-(1-\alpha)^2} \\
&= \frac{\alpha}{2-\alpha}.
\end{aligned} \tag{30}$$

Thus, the variance of $z_i$ is constant.

3. AUTOCORRELATION DEPENDS ONLY ON TIME LAG

The autocorrelation function of $z_i$ is defined as:

$$R_z(\tau) = \mathbb{E}[z_i z_{i+\tau}]. \tag{31}$$

Substituting $z_i = \sum_{k=0}^{\infty} \alpha(1-\alpha)^k \cdot x_{i-k}$ and $z_{i+\tau} = \sum_{j=0}^{\infty} \alpha(1-\alpha)^j \cdot x_{i+\tau-j}$:

$$\begin{aligned}
R_z(\tau) = \mathbb{E}\Bigg[ &\left( \sum_{k=0}^{\infty} \alpha(1-\alpha)^k \cdot x_{i-k} \right) \cdot \\
&\left( \sum_{j=0}^{\infty} \alpha(1-\alpha)^j \cdot x_{i+\tau-j} \right) \Bigg].
\end{aligned} \tag{32}$$

Expanding the product:

$$R_z(\tau) = \sum_{k=0}^{\infty} \sum_{j=0}^{\infty} \alpha^2 (1-\alpha)^{k+j} \cdot \mathbb{E}[x_{i-k} x_{t+\tau-j}]. \tag{33}$$

Since $y(t)$ is zero-mean and uncorrelated at different time points:

$$\mathbb{E}[x_{i-k} z_{i+\tau-j}] = \begin{cases} 1 & \text{if } i-k = i+\tau-j \\ 0 & \text{otherwise.} \end{cases} \tag{34}$$

This implies that the autocorrelation depends only on the time lag $\tau$, not on the specific time $t$. Thus, $R_z(\tau)$ satisfies the condition for stationarity.

See the specific algorithm in Algorithm 1.

## D    COMPARISON BETWEEN PHYSLLM, CNN-LLM HYBRID, AND TRANSFORMER BASED METHOD

We supplement our study with a comparative experiment between PhysLLM and alternative approaches in Fig. 6. First, we compare with a CNN-LLM Hybrid approach, which refers to utilizing the same pretrained PhysNet encoder to extract features, which are then fed into the LLM for further

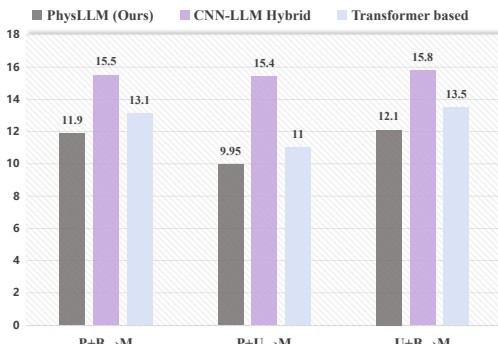 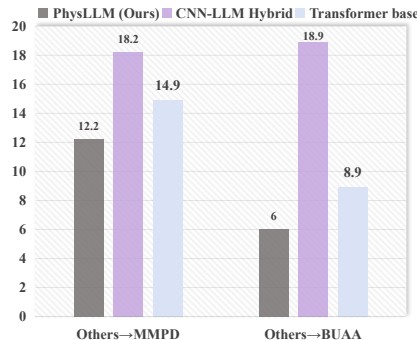

Figure 6: Comparison between PhysLLM, CNN-LLM Hybrid, and Transformer based method. The lower the MAE, the better the effect.

analysis. We adopt the same cross-domain evaluation strategy as in the main text, following both the Two-training and One-testing protocol and the Three-training and One-testing protocol. The experimental results demonstrate that our approach significantly outperforms the direct combination method and also characterize the effective integration of our methods.

Furthermore, we investigate the effectiveness of using LLM architecture compared to a Transformer-based approach. As shown in Fig. 6, while the Transformer-based method achieves competitive performance, our PhysLLM still demonstrates superior results across different evaluation protocols. This comparison validates our architectural choice of using LLM over simpler Transformer models, as the more sophisticated LLM structure better captures the complex physiological patterns and their relationships in the data, leading to more robust and accurate predictions in cross-domain scenarios.

## E    EXAMPLE OF GENERATED VISION CUE

To validate the effectiveness of the generated vision cues, we compare the prompts generated by LLaVA-based question answering (Liu et al., 2023a) in our method with manually crafted prompts to investigate whether they correctly capture the knowledge required for the rPPG task. As shown in Fig. 10, we select four representative video frames, covering scenarios such as dim lighting conditions, head movements, facial occlusions due to beards and glasses, among others. From the prompts, it can be observed that the generated prompts effectively capture the desired details, such as gender, potential head orientation, background color, and accessories. Moreover, the generated prompts exhibit greater granularity; for instance, in the generated prompt of (a) in Fig. 10, they explicitly describe differences in lighting conditions. This comparison demonstrates the rationality and effectiveness of our question-answering approach.

Furthermore, we visualize the BVP signals obtained from PhysLLM under the guidance of both the generated and manually crafted cues. The results indicate that prompts with finer details lead to more stable signals, further validating the effectiveness of our designed cue-guided prompting strategy.

## F    VISUALIZATION OF THE PREDICTED AND GROUND-TRUTH BVP AND PSD

We randomly select clip samples from UBFC-rPPG (Bobbia et al., 2019) and PURE (Stricker et al., 2014) and plot the predicted rPPG and the corresponding PSD signals in Fig. 7. The results clearly demonstrate that PhysLLM effectively predicts the rPPG signals across different datasets and outputs the corresponding smooth waveform.

## G    ADDITIONAL ABLATION STUDY

**Impact of the Hyperparameters.**    We have described all the configurations in Section 3, including the parameter of DDS and the length of the learnable prompt. These hyperparameters are selected based on experience. To verify that our chosen configuration for PhysLLM is suitable, we conduct ablation studies on the parameters $\alpha$ in the DDS, the length of each learnable prompt. The results are shown in Fig. 8. Note that only one parameter is changed at one time, while the others remain unchanged.

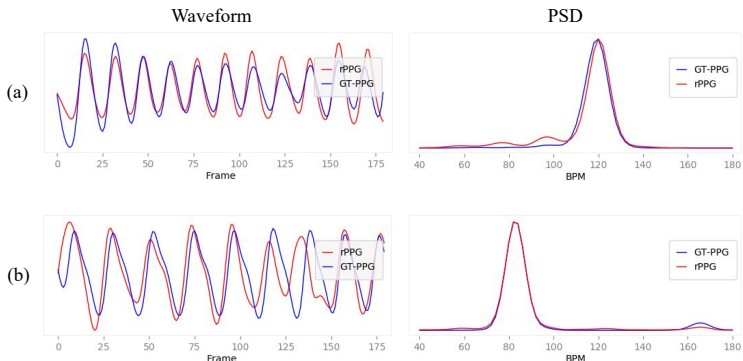

Figure 7: Visual comparison of the rPPG signals (left) predicted by PhysLLM and their corresponding PSDs (right), alongside the respective ground-truth. (a) UBFC-rPPG (Bobbia et al., 2019), (b) PURE (Stricker et al., 2014).

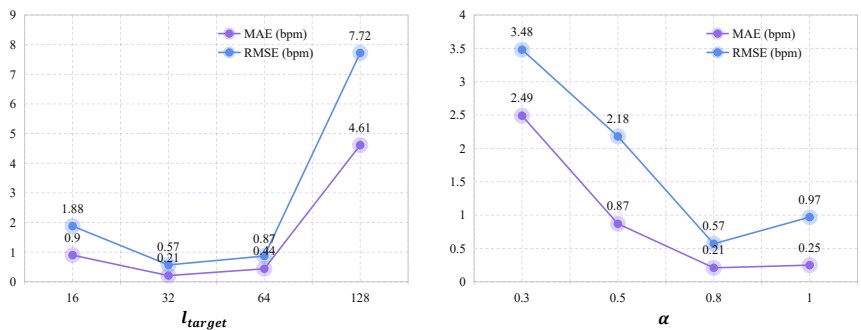

Figure 8: Ablation study of the hyperparameters of $l_{target}$ and $\alpha$ in PhysLLM on UBFC-rPPG (Bobbia et al., 2019) dataset.

**Impact of Different rPPG Backbones.** To verify the generalization of our framework, in addition to PhysNet (Yu et al., 2019) as the backbone, we also consider two other backbones, namely PhysFormer (Yu et al., 2022) and EfficientPhys (Liu et al., 2021a). As shown in Fig. 9, the integration of PhysLLM consistently reduces the RMSE across all backbone architectures and datasets, highlighting its effectiveness. Specifi-

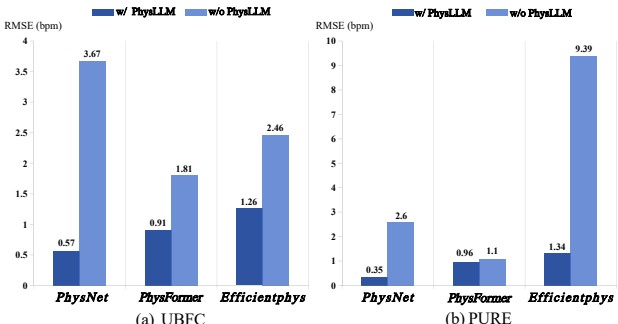

Figure 9: Impact of different rPPG backbones.

cally, on the UBFC dataset (Fig.9(a)), PhysLLM reduces the RMSE from 3.67 bpm to 0.57 bpm for PhysNet, from 1.81 bpm to 0.91 bpm for PhysFormer, and from 2.46 bpm to 1.26 bpm for EfficientPhys. A similar trend is observed on the PURE dataset (Fig.9(b)), where PhysLLM lowers the RMSE from 2.6 bpm to 0.35 bpm for PhysNet, from 1.1 bpm to 0.96 bpm for PhysFormer, and from 9.39 bpm to 1.34 bpm for EfficientPhys. These results clearly demonstrate that PhysLLM serves as a robust enhancement module that generalizes well across different backbone designs and data distributions. Notably, the substantial improvements on EfficientPhys, particularly on the PURE dataset, suggest that PhysLLM can compensate for weaker backbone performance and enhance reliability in more challenging scenarios.

## H    ADDITIONAL RESULTS ON RESPIRATORY RATE PREDICTION

As shown in Table 9, we conduct a comprehensive comparison of respiratory rate (RR) prediction performance on three benchmark datasets: UBFC, PURE, and MMPD. The goal of this evaluation

Table 9: Comparison of MAE and RMSE for respiratory rate estimation on the UBFC, PURE, and MMPD datasets.

| Method | UBFC | | PURE | | MMPD | |
|---|---|---|---|---|---|---|
| | MAE | RMSE | MAE | RMSE | MAE | RMSE |
| PhysNet (Yu et al., 2019) | 15.82 | 17.84 | 13.78 | 16.46 | 10.30 | 13.70 |
| PhysFormer (Yu et al., 2022) | 6.15 | 9.87 | 11.37 | 14.73 | 9.91 | 13.79 |
| EfficientPhys (Liu et al., 2021a) | 9.59 | 13.06 | 8.71 | 12.13 | 11.97 | 14.57 |
| RhythmFormer (Zou et al., 2025) | 4.16 | 7.89 | 7.83 | 11.89 | **6.37** | **8.89** |
| PhysLLM (Ours) | **4.05** | **7.70** | **6.66** | **9.32** | 7.03 | 11.38 |

is to assess the generalizability and robustness of our method across diverse subject domains and recording conditions.

Our approach consistently outperforms prior state-of-the-art methods, including PhysNet, Phys-Former, and EfficientPhys, across all datasets. In particular, we observe a substantial reduction in error on the UBFC dataset, where our method achieves an MAE of 4.05 and RMSE of 7.70. Similarly, our method maintains strong performance on the PURE and MMPD datasets, outperforming others by a large margin.

These results demonstrate the effectiveness of our model in extracting reliable respiratory signals from facial videos and generalizing to different datasets. The improvements are attributed to our model's robust spatiotemporal representation and domain-invariant design, which allow it to maintain accuracy even under varying motion, lighting, and subject conditions.

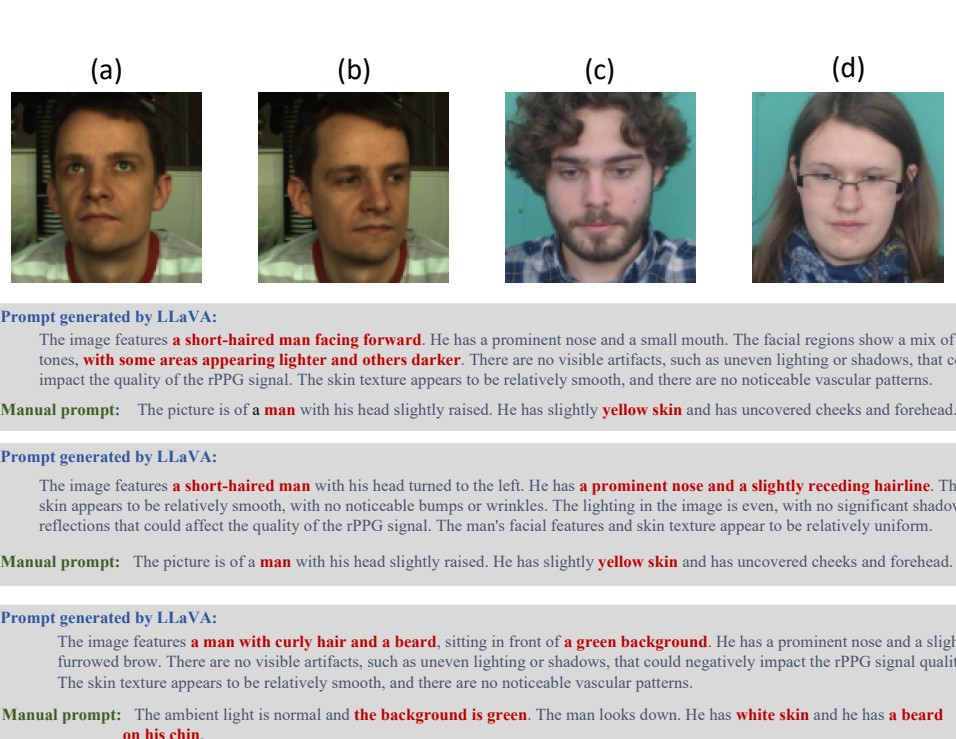

**(a) Prompt generated by LLaVA:**
The image features **a short-haired man facing forward**. He has a prominent nose and a small mouth. The facial regions show a mix of skin tones, **with some areas appearing lighter and others darker**. There are no visible artifacts, such as uneven lighting or shadows, that could impact the quality of the rPPG signal. The skin texture appears to be relatively smooth, and there are no noticeable vascular patterns.

**Manual prompt:** The picture is of a **man** with his head slightly raised. He has slightly **yellow skin** and has uncovered cheeks and forehead.

**(b) Prompt generated by LLaVA:**
The image features **a short-haired man** with his head turned to the left. He has **a prominent nose and a slightly receding hairline**. The man's skin appears to be relatively smooth, with no noticeable bumps or wrinkles. The lighting in the image is even, with no significant shadows or reflections that could affect the quality of the rPPG signal. The man's facial features and skin texture appear to be relatively uniform.

**Manual prompt:** The picture is of a **man** with his head slightly raised. He has slightly **yellow skin** and has uncovered cheeks and forehead.

**(c) Prompt generated by LLaVA:**
The image features **a man with curly hair and a beard**, sitting in front of **a green background**. He has a prominent nose and a slightly furrowed brow. There are no visible artifacts, such as uneven lighting or shadows, that could negatively impact the rPPG signal quality. The skin texture appears to be relatively smooth, and there are no noticeable vascular patterns.

**Manual prompt:** The ambient light is normal and **the background is green**. The man looks down. He has **white skin** and he has **a beard on his chin**.

**(d) Prompt generated by LLaVA:**
The image features **a woman wearing glasses and a scarf**, sitting in front of **a green background**. She has a prominent nose and a slightly pudgy face. The facial regions show a mix of skin tones, with the forehead being lighter than the cheeks and nose. The skin appears to be relatively smooth, with no visible vascular patterns or significant texture.

**Manual prompt:** The ambient light is normal and the background is **green**. The **woman looks down**. She has whiter skin and her cheek, forehead is uncovered.

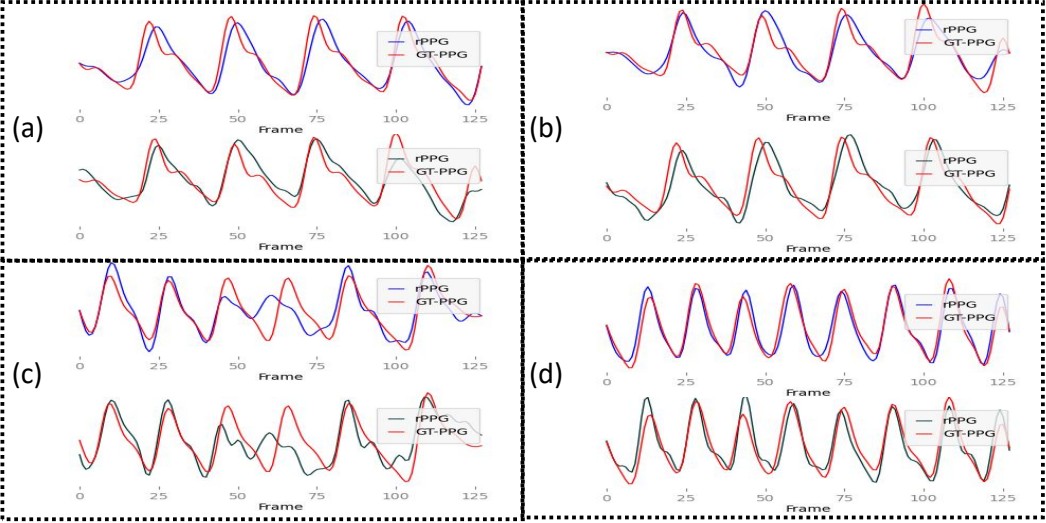

Figure 10: Comparison between manual prompt and prompt generated by LLaVA (Liu et al., 2023a). The top section presents the selected video frames, the middle section compares the generated vision cues with manually crafted prompts, and the bottom section illustrates the BVP signal predictions from PhysLLM, guided by the two types of cues.

