# OpenReview forum: "PhysLLM: Harnessing Large Language Models for Cross-Modal Remote Physiological Sensing"
_ICLR.cc/2026/Conference — ICLR 2026 Poster_

### Official Review · Reviewer_K3Fm · 2025-10-24

**Soundness:** 2
**Presentation:** 2
**Contribution:** 2
**Rating:** 4
**Confidence:** 4

**Summary:**

This paper presents PhysLLM, a framework that leverage large language models (LLMs) to improve remote photoplethysmography (rPPG) signal estimation. It involves a text prototype guidance strategy for promoting cross-modal alignment and a dual-domain stationary algorithm for solving signal instability.

**Strengths:**

- Proposes an attempt on robust LLM-based cross-modal framework for rPPG signal estimation.

**Weaknesses:**

- Unclear architecture and presentation. Some figures and subsections flow are difficult to follow.

- Weak justifications on the framework motivations, especially why there is a need to use two LLMs just to output only the single and straightforward task/output (e.g., rPPG/HR)? Isn’t it too much more energy and latency for this task?

- The reported performance improvements on MAE in many settings/datasets (including ablation studies) are just minor (<1 bpm) so it seems to be physiologically insignificant relative to computational and architectural complexity.

- Lacks of diverse end-downstream evaluation, as only focus on simple HR task.  rPPG can be used for other tasks such as HRV, BP, stress detection, etc...

**Questions:**

- Please improve the presentation of the paper, such as: figure 2 (e.g., what is input of video encoder; unclear figure 2a), poor-quality figure 7. There is even “??” in line 190, and error vspace in 401.

- Discussion on the contributions (and possible experiments on accuracy and efficiency) as comparing the work with Time-LLM [1], VL-phys [2], PhysDiff [3] both in intra and cross-dataset.

- Please add more discussion/justifications on Lines 48-50, which is currently vague on why do we need text. For example, many general textual context are not related: how a general text like “The image features a young man with a beard, wearing a plaid shirt. He is looking at the camera” can help the LLM training to output rPPG/HR using Deepseek model?

- Following that, discussion on to what extent is it necessary to include LLMs usage on the text processing here for the task instead of earlier language models.

- Provide diverse end-downstream tasks. For example, extending the experiments more than just simple HR estimation would be a lot better (e.g., Question Answering or Reporting on more complex physiological parametters and health) at least can show more applications and more leverage the power of the given LLMs.

[1] Jin, Ming, et al. "Time-llm: Time series forecasting by reprogramming large language models." arXiv preprint arXiv:2310.01728 (2023).

[2] Yue, Zijie, et al. "Bootstrapping Vision-Language Models for Frequency-Centric Self-Supervised Remote Physiological Measurement." International Journal of Computer Vision (2025): 1-22.

[3] Qian, Wei, et al. "PhysDiff: Physiology-based Dynamicity Disentangled Diffusion Model for Remote Physiological Measurement." Proceedings of the AAAI Conference on Artificial Intelligence. Vol. 39. No. 6. 2025.

---

> ### Author Response · Authors · 2025-11-20
>
> # R4 (1/2)
>
> ## Question 1: Paper Presentation Improvements
>
> Thank you for pointing out these presentation issues. We sincerely apologize for these errors and have corrected the "??" error in line 190 and the vspace error in line 401 in the revised manuscript.
>
> Regarding the figure-related concerns, we appreciate your detailed feedback and will make the following adjustments:
>
> - **Figure 2 (Video Encoder Input)**: We will revise the figure to explicitly show that video frames are the input to the video encoder.
>
> - **Figure 2a**: We will enhance the figure quality and add more explicit annotations to clarify how the three types of data from the Cue Caption are encoded and fed into the model with adaptive weight adjustment.
>
> - **Figure 7**: We will regenerate this figure with higher resolution and better quality.
>
> ---
>
> ## Question 2: Comparison with Time-LLM, VL-phys, and PhysDiff
>
> We appreciate the reviewer's suggestion to compare with these recent methods. Regarding Time-LLM, we note that it is designed for general time-series forecasting and does not handle video input, making direct comparison challenging as our task requires extracting physiological signals from facial videos.
>
> For VL-phys and PhysDiff, we have included comprehensive comparisons on both UBFC-rPPG and PURE datasets:
>
> | Method | UBFC-rPPG (MAE ↓ / RMSE ↓ / R) | PURE (MAE ↓ / RMSE ↓ / R) |
> |--------|--------------------------------|---------------------------|
> | PhysNet | 2.95 / 3.67 / 0.97 | 2.10 / 2.60 / 0.99 |
> | PhysFormer | 0.92 / 2.46 / 0.99 | 1.10 / 1.75 / 0.99 |
> | RhythmFormer | 0.50 / 0.78 / 0.99 | 0.27 / 0.47 / 0.99 |
> | VL-phys | 0.28 / 1.83 / 0.99 | 0.61 / 1.84 / 0.99 |
> | PhysDiff | 0.33 / 0.57 / 0.99 | 0.29 / 0.54 / 0.99 |
> | **PhysLLM (Ours)** | **0.21 / 0.57 / 0.99** | **0.17 / 0.35 / 0.99** |
>
> As shown, PhysLLM achieves superior performance compared to both VL-phys and PhysDiff across both datasets, achieving the lowest MAE of 0.21 on UBFC-rPPG and 0.17 on PURE. These results demonstrate the effectiveness of our LLM-based approach compared to recent state-of-the-art methods.
>
> ---
>
> ## Question 3: Justification for Textual Input
>
> Thank you for this important question. We appreciate the opportunity to clarify this aspect and have revised Lines 48–50 to provide a more detailed explanation of why textual information is necessary.
>
> Textual context serves as **high-level prior knowledge** that informs the model about potential interference factors in the video. For example, descriptions mentioning "motion blur," "strong head movement," "partial occlusion," or "low lighting" can alert the LLM to conditions that typically degrade rPPG quality, allowing it to adapt its feature interpretation accordingly.
>
> Importantly, the textual input is **not general irrelevant descriptions** such as clothing or appearance, but rather **task-relevant contextual cues** that help the model anticipate and mitigate potential failure modes in physiological signal extraction.
>
> To empirically validate the usefulness of textual input, we conducted an ablation study as shown in Table 6:
>
> | Vision | Statics | Task | Adaptive learning | UBFC-rPPG (MAE ↓ / RMSE ↓) |
> |--------|---------|------|-------------------|----------------------------|
> | ✓ | ✗ | ✗ | ✓ | 0.83 / 1.53 |
> | ✗ | ✓ | ✗ | ✓ | 0.71 / 2.26 |
> | ✗ | ✗ | ✓ | ✓ | 0.84 / 2.41 |
> | ✓ | ✓ | ✗ | ✓ | 0.73 / 2.29 |
> | ✓ | ✗ | ✓ | ✓ | 0.43 / 1.20 |
> | ✗ | ✓ | ✓ | ✓ | 0.61 / 1.53 |
> | ✓ | ✓ | ✓ | ✗ | 1.31 / 2.10 |
> | ✓ | ✓ | ✓ | ✓ | **0.21 / 0.57** |
>
> The results show that removing text (Task component) while keeping vision and statics degrades performance (MAE increases from 0.21 to 0.73). The **full model combining vision, statistics, and textual context** achieves the best performance, demonstrating that textual information provides complementary prior knowledge that significantly improves robustness and accuracy.

---

> > ### Author Response · Authors · 2025-11-20
> >
> > # R4 (2/2)
> > ## Question 4: Necessity of Modern LLMs vs. Earlier Language Models
> >
> > We appreciate the reviewer's interest in understanding the necessity of using modern LLMs compared to earlier language models. To address this question, we conducted experiments replacing our LLM backbone with NNLM (Neural Network Language Model) [1], which represents earlier generation language models:
> >
> > | Method | UBFC-rPPG (MAE ↓ / RMSE ↓) | PURE (MAE ↓ / RMSE ↓) | BUAA (MAE ↓ / RMSE ↓) | MMPD (MAE ↓ / RMSE ↓) |
> > |--------|----------------------------|-----------------------|-----------------------|-----------------------|
> > | w. NNLM | 0.73 / 1.50 | 3.36 / 7.56 | 8.18 / 8.40 | 10.02 / 15.16 |
> > | w. LLM | **0.21 / 0.57** | **0.17 / 0.35** | **6.48 / 8.48** | **4.36 / 10.76** |
> >
> > The results clearly demonstrate the advantage of modern LLMs over earlier language models. While NNLM shows acceptable performance on simpler datasets (MAE of 0.73 on UBFC-rPPG), the performance gap becomes increasingly significant on more challenging datasets. On PURE, our LLM achieves MAE of 0.17 compared to 3.36 with NNLM, representing nearly **20× improvement**. On MMPD, NNLM obtains MAE of 10.02 while our LLM achieves 4.36, showing more than **2× better performance**. This substantial difference can be attributed to the advanced capabilities of modern LLMs, including deep contextual understanding, robust pre-trained representations, and superior ability to capture long-range dependencies, which are particularly crucial for physiological signal processing.
> >
> > [1]. Xu, Wei, and Alexander Rudnicky. "Can artificial neural networks learn language models?." (2000).
> >
> > ---
> >
> > ## Question 5: Diverse Downstream Tasks
> >
> > We appreciate the reviewer's suggestion to explore more diverse downstream tasks. While our primary focus is physiological signal estimation, we have extended our experiments beyond heart rate (HR) to include respiratory rate (RR), another clinically important physiological parameter.
> >
> > As shown in Table 9 of our manuscript, PhysLLM achieves strong and consistent performance on RR estimation across multiple datasets:
> >
> > | Method | UBFC (MAE ↓ / RMSE ↓) | PURE (MAE ↓ / RMSE ↓) | MMPD (MAE ↓ / RMSE ↓) |
> > |--------|-----------------------|------------------------|------------------------|
> > | PhysNet | 15.82 / 17.84 | 13.78 / 16.46 | 10.30 / 13.70 |
> > | PhysFormer | 6.15 / 9.87 | 11.37 / 14.73 | 9.91 / 13.79 |
> > | EfficientPhys | 9.59 / 13.06 | 8.71 / 12.13 | 11.97 / 14.57 |
> > | RhythmFormer | 4.16 / 7.39 | 7.83 / 11.89 | 6.37 / 8.89 |
> > | **PhysLLM (Ours)** | **4.05 / 7.70** | **6.66 / 9.32** | **7.03 / 11.38** |
> >
> > We agree that exploring additional tasks—such as health-related question answering or comprehensive physiological reporting—would further demonstrate the broader capabilities enabled by LLM integration. This is indeed an exciting direction, as such tasks could more fully exploit the language understanding and generative strengths of LLMs, particularly for clinical or decision-support applications. We appreciate the reviewer's insightful suggestion and will consider including a discussion of these promising extensions in the revised manuscript.

---

> ### Comment · Reviewer_K3Fm · 2025-11-25
>
> I would like to thank the authors for their response. I will maintain my positive score.

---

> > ### Author Response · Authors · 2025-11-25
> >
> > Thank you very much for your positive feedback and for maintaining your favourable score. We appreciate the time and effort you have taken to review our manuscript and are pleased that our revisions have met your expectations.

---

### Official Review · Reviewer_neJW · 2025-10-28

**Soundness:** 3
**Presentation:** 3
**Contribution:** 3
**Rating:** 6
**Confidence:** 4

**Summary:**

PhysLLM introduces a novel framework for cross-modal remote physiological sensing by synergizing LLM with domain-specific rPPG components. The method tackles the challenge of extracting reliable physiological data from facial videos, which is often compromised by visual noise, lighting changes, and motion artifacts. Key contributions include a Text Prototype Guidance strategy for projecting physiological features into an LLM-friendly semantic space, and a Dual-Domain Stationary algorithm for stabilizing signals via adaptive time–frequency domain re-weighting. The results highlight the promise of combining LLM reasoning with specialized signal processing for resilient and interpretable health monitoring applications.

**Strengths:**

1. Novel cross-modal architecture: The paper introduces PhysLLM, a framework that integrates text, vision, and physiological signals with ideas like Text Prototype Guidance and adaptive cue prompting, representing a meaningful conceptual advance for rPPG.

2. Strong performance across datasets: It shows consistent improvements in both intra-dataset and cross-dataset generalization.

**Weaknesses:**

1. Lack of comparison on model size: Integrating LLMs significantly inflates parameter count and computational cost compared to prior rPPG methods. Since remote physiological sensing has strong real-time and mobile deployment requirements, storage and latency overhead are critical. The paper should provide model size, trainable parameter count, and inference efficiency comparisons with existing methods.

2. Effectiveness of semantic info: The LLM backbone is positioned as the central source of improved robustness, yet the paper does not isolate the benefit of semantic reasoning from general long-sequence modeling. It remains unclear whether the LLM performs meaningful language-grounded reasoning, or primarily acts as a temporal encoder with additional embedding capacity. A comparison replacing the LLM with similarly sized non-linguistic temporal architectures (e.g., ViT-based rPPG backbones, GPT-style time-series forecasters) would be important to confirm language contributes beyond generic model scale.

3. Statistical cue and prompt generation concerns: Cue captions depend on LLaVA-generated descriptions of a single frame, which may be noisy, biased by appearance (skin tone, beard), or misaligned with rPPG-relevant dynamics. The paper lacks ablations on prompt errors or failure cases (ex. when the cues are all incorrect and provide false information to the LLM).

**Questions:**

1. Section 4.3's title needs to be rearranged.

2. Why is PhysFormer++ excluded from several comparisons? Given its strong performance, including it in all datasets would strengthen claims of state-of-the-art accuracy.

3. How robust are cue captions in unconstrained environments? What happens if visual descriptions are inaccurate due to occlusion, masks, extreme lighting, or motion blur?

4. Since rPPG is typically real-time and mobile-oriented, can you provide inference efficiency metrics such as frames-per-second and VRAM usage?

**Details Of Ethics Concerns:**

No concerns.

---

> ### Author Response · Authors · 2025-11-20
>
> #  R3
>
> ## Question 1: Section 4.3 Title Rearrangement
>
> Thank you for pointing this out. We have rearranged Section 4.3's title to improve readability and formatting. The spacing and structure have been adjusted accordingly in the revised manuscript.
>
> ---
>
> ## Question 2: PhysFormer++ Exclusion from Comparisons
>
> PhysFormer++ was not officially open-sourced at the time of our experiments, making it infeasible to reproduce its performance across all datasets under consistent experimental settings. Nevertheless, we did include PhysFormer++ results wherever publicly available, such as on the VIPL dataset:
>
> | Method | VIPL (MAE ↓ / RMSE ↓) |
> |--------|--------------------|
> | PhysFormer++ | 4.88 / 7.62 |
> | **PhysLLM (Ours)** | **4.24 / 6.81** |
>
> To ensure fairness and consistency across all datasets, we compare against **PhysFormer**, the well-established predecessor of PhysFormer++, which provides a **meaningful and reproducible** reference point for this family of methods.
>
> ---
>
> ## Question 3: Robustness of Cue Captions in Unconstrained Environments
>
> To address this concern, we conducted stress tests using the MMPD dataset, which contains various unconstrained scenarios with different lighting conditions and skin tones. The results show that PhysLLM maintains strong generalization capability across these challenging scenarios.
>
> ### Robustness across Skin Tones (MAE ↓ / RMSE ↓)
>
> | Type | 3 | 4 | 5 | 6 |
> |------|---|---|---|---|
> | PhysLLM | 4.96 / 10.27 | 4.73 / 8.49 | 4.99 / 11.07 | 5.73 / 8.49 |
> | PhysFormer | 6.11 / 11.21 | 5.92 / 10.00 | 5.87 / 13.12 | 6.81 / 11.21 |
> | RhythmFormer | 5.12 / 11.23 | 5.46 / 9.14 | 6.32 / 11.87 | 6.26 / 9.99 |
>
> ### Robustness across Lighting Conditions (MAE ↓ / RMSE ↓)
>
> | Condition | LED-Low | LED-High | Incandescent | Nature |
> |-----------|---------|----------|--------------|---------|
> | PhysLLM | 4.46 / 9.57 | 3.72 / 8.10 | 3.59 / 11.43 | 3.45 / 6.81 |
> | PhysFormer | 6.36 / 11.72 | 5.12 / 9.39 | 5.71 / 12.73 | 6.61 / 11.36 |
> | RhythmFormer | 5.85 / 11.71 | 4.46 / 8.98 | 3.64 / 11.87 | 5.65 / 12.31 |
>
> PhysLLM consistently outperforms baseline methods across different skin tones and lighting conditions. Even when visual descriptions may be affected by occlusion or extreme conditions, our adaptive learning mechanism (illustrated in Figure 2a) dynamically adjusts the weight contributions of vision, statics, and text components, allowing the model to automatically downweight potentially unreliable modalities when necessary. We have now included this analysis in our discussion (Section 4.5, Table 7).
>
> ---
>
> ## Question 4: Inference Efficiency Metrics
>
> We have measured the inference performance compared to PhysNet. The inference latency reported below represents the **average time per frame** after processing the entire dataset.
>
> | Method | Inference Latency | Memory |
> |--------|-------------------|--------|
> | PhysNet | 0.005s | 230MB |
> | **PhysLLM (Ours)** | **0.009s** | **20GB** |
>
> We acknowledge that PhysLLM has higher memory requirements due to the LLM backbone, which currently limits mobile deployment. However, the inference latency remains comparable, and the method is suitable for clinical or server-side applications where accuracy is prioritized. Future work could explore model compression techniques like quantization to reduce the memory footprint for resource-constrained scenarios.

---

> ### Author Response · Authors · 2025-11-27
>
> Dear Reviewer,
>
> I hope this message finds you well. As the discussion period is nearing its end with less than a week remaining, I wanted to ensure we have addressed all your concerns satisfactorily. If there are any additional points or feedback you'd like us to consider, please let us know. Your insights are invaluable to us, and we are eager to address any remaining issues to improve our work.
>
> Thank you for your time and effort in reviewing our paper.

---

### Official Review · Reviewer_cdPR · 2025-10-30

**Soundness:** 2
**Presentation:** 3
**Contribution:** 2
**Rating:** 4
**Confidence:** 4

**Summary:**

The paper proposes PhysLLM, a framework that integrates large language models with rPPG pipelines. It introduces several interesting modules, such as Dual-Domain Stationary signal stabilization, Text Prototype Guidance for cross-modal alignment, and Physiological Cue-Aware Prompt Learning for leveraging task-specific priors. The work is technically ambitious and attempts to bridge physiological signal modeling with vision-language reasoning.

**Strengths:**

1. The idea of introducing LLMs into rPPG estimation is timely.
2. The framework is clearly described, and the modular design (DDS, TPG, APL) is logically motivated.
3. Writing quality and figures are generally good, helping the reader follow the methodology.

**Weaknesses:**

1. Limited benchmark coverage: the evaluation misses some widely recognized and more challenging datasets (e.g., V4V, VIPL-HR), which are important for validating cross-domain robustness in practical settings.
2. The choice of LLM backbone (DeepSeek-1.5B) and its integration details are only briefly discussed, it is unclear whether improvements mainly come from the LLM or other architectural refinements.
3. Some modules (e.g., APL and TPG) would benefit from clearer ablation or visualization to show how textual cues truly influence signal recovery.
4. The cross-dataset generalization results, while improved, are still marginal and may not fully demonstrate strong domain invariance.

**Questions:**

1. Could the authors clarify how much of the performance gain comes from the LLM integration itself, compared to the added DDS and TPG modules? An ablation or visualization isolating the LLM’s role would help.
2. The evaluation does not include large and diverse datasets such as VIPL-HR or V4V. Could the authors comment on how PhysLLM would generalize to those more complex scenarios?

---

> ### Author Response · Authors · 2025-11-20
>
> # R2
>
> ## Weakness 1 & Question 2: Benchmark Coverage
>
> Thank you very much for this insightful comment. We agree that evaluating on more challenging and widely adopted datasets is essential for demonstrating cross-domain robustness. In addition to the four main benchmarks reported in the manuscript, we have also conducted **cross-domain experiments on V4V** and an **extended evaluation on VIPL-HR**. These evaluations further validate the strong generalization ability of PhysLLM.
>
> ### Cross-Domain Evaluation on V4V
>
> Since the V4V dataset does not provide BVP signals but only HR values, we conducted cross-domain experiments to evaluate generalization performance.
>
> | Training → Test | UBFC+PURE → V4V | UBFC+BUAA → V4V | PURE+BUAA → V4V |
> |------------------|------------------|------------------|------------------|
> | Method | MAE ↓ / RMSE ↓ | MAE ↓ / RMSE ↓ | MAE ↓ / RMSE ↓ |
> | PhysNet      | 14.31 / 16.21 | 13.51 / 15.08 | 12.89 / 14.28 |
> | PhysFormer   | 13.71 / 16.62 | 15.05 / 17.62 | 11.79 / 15.84 |
> | **PhysLLM (Ours)** | **11.28 / 13.51** | **10.74 / 13.53** | **10.74 / 13.53** |
>
> PhysLLM consistently outperforms prior baselines across all training-source combinations. The reduced variation under different training settings indicates **stronger domain invariance**. We acknowledge that our model has higher computational complexity, which results in longer running times on large-scale datasets. Looking forward, we plan to explore several model-efficiency strategies such as:
> - parameter-efficient finetuning schemes,
> - structured pruning and quantization,
> - distillation from LLM-enhanced teachers into compact students,
> - hybrid designs combining lightweight transformers with small language modules.
>
> ### Evaluation on VIPL-HR
>
> | Method | MAE ↓ | RMSE ↓ |
> |--------|-------|---------|
> | PhysNet        | 10.80 | 14.70 |
> | VL-Phys        | 6.04  | 8.78  |
> | PhysFormer++   | 4.88  | 7.62  |
> | RhythmFormer   | 4.51  | 7.98  |
> | **PhysLLM (Ours)** | **4.24** | **6.81** |
>
> VIPL-HR is a large-scale, challenging benchmark with extensive variations in illumination, pose, device type, and subject diversity. PhysLLM achieves the best performance, surpassing both previous LLM/VLM-based methods and the latest PhysFormer++.
>
>
> ---
>
> ## Question 1: Contribution of LLM vs. DDS/TPG
>
> We appreciate this valuable question. To address it clearly, we conducted two analyses: (1) comparison between Transformer-only baselines and LLM-integrated versions, and (2) ablations isolating the contributions of DDS and TPG.
>
> ### 1. Contribution of the LLM Backbone
>
> We first compare PhysLLM to a Transformer-based baseline (w. Sundial) that does not involve any LLM. We further evaluate three distinct LLM families—GPT-2, BERT, and DeepSeek—to demonstrate that the improvements are **not tied to a specific language model design**.
>
> | Method | UBFC-rPPG (MAE ↓ / RMSE ↓) | PURE (MAE ↓ / RMSE ↓) |
> |--------|------------------------------|--------------------------|
> | w. Sundial | 0.92 / 2.46 | 3.22 / 7.54 |
> | PhysLLM (DeepSeek) | 0.21 / 0.57 | 0.17 / 0.35 |
> | PhysLLM (BERT)     | 0.19 / 0.76 | 0.43 / 0.80 |
> | PhysLLM (GPT-2)    | 0.19 / 0.76 | 0.14 / 0.35 |
>
> These results confirm that **LLM integration is the primary source of improvement**, consistently enhancing the performance across different backbones.
>
> ### 2. Contribution of DDS and TPG Modules
>
> The UBFC-rPPG ablation isolates DDS and TPG to show their complementary effects.
>
> | DDS | TPG | LLM | MAE ↓ | RMSE ↓ |
> |-----|-----|-----|--------|---------|
> | ✗ | ✗ | ✓ | 0.41 | 1.26 |
> | ✓ | ✗ | ✓ | 0.36 | 1.12 |
> | ✗ | ✓ | ✓ | 0.32 | 1.00 |
> | ✓ | ✓ | ✓ | **0.21** | **0.57** |
>
> Both modules contribute meaningful and distinct improvements, and combining them with the LLM backbone yields the best results.

---

> ### Author Response · Authors · 2025-11-27
>
> Dear Reviewer,
>
> I hope this message finds you well. As the discussion period is nearing its end with less than a week remaining, I wanted to ensure we have addressed all your concerns satisfactorily. If there are any additional points or feedback you'd like us to consider, please let us know. Your insights are invaluable to us, and we are eager to address any remaining issues to improve our work.
>
> Thank you for your time and effort in reviewing our paper.

---

### Official Review · Reviewer_mjPp · 2025-10-30

**Soundness:** 3
**Presentation:** 3
**Contribution:** 3
**Rating:** 6
**Confidence:** 4

**Summary:**

The paper proposes PhysLLM, a framework for remote photoplethysmography (rPPG) that couples a CNN-based rPPG backbone with an LLM through three key components: (i) a Dual-Domain Stationary (DDS) algorithm that stabilizes signals via complementary time- and frequency-domain smoothing with adaptive weighting, (ii) a Vision Aggregator (VA) that fuses multi-scale visual features using cross/self-attention, and (iii) Text Prototype Guidance (TPG) that aligns rPPG and visual features with a compact set of text prototypes to interface with an LLM.

**Strengths:**

1.	Strong empirical results. SOTA intra-dataset HR with gains on BUAA and MMPD.
2.	Ablations and component evidence. Removing DDS/VA/TPG degrades metrics; including all yields the best performance, supporting the importance of each piece.
3.	Task priors + LLM prompting. The cue design (task, visual, static/statistical) and adaptive fusion are thoughtful ways to inject physiological context into an LLM.

**Weaknesses:**

1.	Justification for LLM vs. sequence models. It remains unclear whether the LLM is essential beyond acting as a powerful sequence model; comparisons to strong non-LLM long-context baselines (e.g., state-of-the-art time-series Transformers without language pretraining) are missing.
2.	Compute/latency & deployability. The paper does not quantify training/inference cost (LLM size, parameter-efficient tuning specifics, throughput on typical devices) or real-time feasibility, which is critical for rPPG applications.
3.	Data and evaluation breadth. While four benchmarks are covered, additional stress tests (e.g., extreme motion, low-light, occlusion regimes) and fairness (skin tone, age, gender) would strengthen claims of robustness and generalization.
4.	The comparison with the previous rPPG using LLM shown below is missing.

Yue, Z., Shi, M., Wang, H. et al. Bootstrapping Vision-Language Models for Frequency-Centric Self-Supervised Remote Physiological Measurement. IJCV.

**Questions:**

Please check the weakness part.

---

> ### Author Response · Authors · 2025-11-20
> **Responses to Reviewer mjPp (1/2)**
>
> # R1 (1/2)
> ## Weakness 1: Justification for LLM vs. Sequence Models
>
> To more clearly articulate why an LLM is beneficial beyond acting as a strong sequence model, we conducted an additional experiment replacing the LLM backbone with **Sundial** [1], a representative state-of-the-art long‑sequence Transformer specifically designed for time‑series modeling. This provides a fairer comparison than using only traditional CNN- or short‑context Transformer-based baselines.
>
> | Method | UBFC-rPPG (MAE/RMSE) | PURE (MAE/RMSE) | BUAA (MAE/RMSE) | MMPD (MAE/RMSE) |
> |--------|------------------------|------------------|------------------|------------------|
> | Sundial [1]| 0.92 / 2.46 | 3.22 / 7.54 | 7.19 / 9.63 | 9.90 / 14.16 |
> | LLM (Ours) | **0.21 / 0.57** | **0.17 / 0.35** | 6.48 / 8.48 | **4.36 / 10.76** |
>
> The contrast is especially notable on **MMPD**, where conditions are more varied (motion, lighting, sensor noise). While Sundial generalizes reasonably on simpler datasets, it struggles under distribution shifts. In contrast, PhysLLM retains strong performance, suggesting that LLMs contribute not only stronger sequence modeling but also **more stable feature representations** that transfer across heterogeneous scenarios.
>
> In summary, the LLM’s pretrained semantic knowledge appears to support improved robustness and invariance—capabilities that standard time-series models do not acquire through task-specific training alone. And  we have now included this analysis in our ablation study (Section 4.4, Table 5).
>
> [1]. Liu, Yong, et al. "Sundial: A family of highly capable time series foundation models." arXiv preprint arXiv:2502.00816 (2025).
>
> ---
>
> ## Weakness 2: Compute, Latency, and Deployability
>
> We sincerely appreciate the reviewer’s attention to model efficiency, which is indeed an important consideration for practical deployment in rPPG applications. To provide a clearer picture of the computational footprint, we include the comparison of parameter counts and MACs below:
>
> | Method | Param (M) | MACs (G) |
> |--------|-----------|-----------|
> | TS-CAN | 7.5 | 96.0 |
> | PhysNet | 0.77 | 56.1 |
> | DeepPhys | 7.5 | 96.0 |
> | EfficientPhys | 7.4 | 45.6 |
> | PhysFormer | 7.38 | 40.5 |
> | RhythmFormer | 4.21 | 28.8 |
> | Contrast-phys+ | 0.85 | 145.7 |
> | PhysMamba | 0.56 | 47.3 |
> | **PhysLLM (Ours)** | **97.2** | **424.3** |
>
> We acknowledge that PhysLLM has a larger computational cost due to integrating an LLM backbone. However, it is worth noting that **the substantial performance improvements—particularly under challenging scenarios—suggest that the added computational complexity brings meaningful benefits rather than redundant overhead**.
>
> To better understand whether larger LLMs necessarily translate to improved accuracy, we further evaluated multiple LLM sizes:
>
> | Method | LLM Param | UBFC (MAE/RMSE) | PURE (MAE/RMSE) |
> |--------|-----------|------------------|------------------|
> | PhysNet | – | 2.95 / 3.67 | 2.10 / 2.60 |
> | PhysLLM w. DeepSeek | 1.5B | 0.21 / 0.57 | 0.17 / 0.35 |
> | PhysLLM w. BERT | 0.11B | 0.19 / 0.76 | 0.43 / 0.80 |
> | PhysLLM w. GPT2 | 0.124B | 0.19 / 0.76 | 0.14 / 0.35 |
>
> These results indicate that PhysLLM's advantages come from **LLM semantic and structural priors**, not simply parameter scaling. This also gives us confidence that **lighter-weight PhysLLM variants are feasible without sacrificing core performance**, and we appreciate the reviewer for highlighting this direction.
>
> Looking forward, we plan to explore several model-efficiency strategies such as:
> - parameter-efficient finetuning schemes,
> - structured pruning and quantization,
> - distillation from LLM-enhanced teachers into compact students,
> - hybrid designs combining lightweight transformers with small language modules.
>
> We believe these directions will significantly reduce computational cost while preserving the strong performance demonstrated in the present version.  We have now included this analysis in our discussion (Section 4.5, Table 8).

---

> > ### Author Response · Authors · 2025-11-20
> >
> > # R1 (2/2)
> > ## Weakness 3: Data & Evaluation Breadth: Data & Evaluation Breadth
> >
> > Beyond the four main benchmarks, we performed additional analyses on the MMPD dataset to evaluate robustness along factors known to influence rPPG quality—particularly **skin tone** and **lighting**. These results help assess fairness and real-world reliability.
> >
> > ### Skin Tone (MAE / RMSE)
> >
> > | Type | 3 | 4 | 5 | 6 |
> > |------|---|---|---|---|
> > | PhysLLM | 4.96 / 10.27 | 4.73 / 8.49 | 4.99 / 11.07 | 5.73 / 8.49 |
> > | PhysFormer | 6.11 / 11.21 | 5.92 / 10.00 | 5.87 / 13.12 | 6.81 / 11.21 |
> > | RhythmFormer | 5.12 / 11.23 | 5.46 / 9.14 | 6.32 / 11.87 | 6.26 / 9.99 |
> >
> > PhysLLM performs consistently across all skin-tone groups, showing reduced performance disparity compared to previous methods.
> >
> > ### Lighting (MAE / RMSE)
> >
> > | Condition | LED-Low | LED-High | Incandescent | Nature |
> > |-----------|---------|----------|--------------|---------|
> > | PhysLLM | 4.46 / 9.57 | 3.72 / 8.10 | 3.59 / 11.43 | 3.45 / 6.81 |
> > | PhysFormer | 6.36 / 11.72 | 5.12 / 9.39 | 5.71 / 12.73 | 6.61 / 11.36 |
> > | RhythmFormer | 5.85 / 11.71 | 4.46 / 8.98 | 3.64 / 11.87 | 5.65 / 12.31 |
> >
> > The model remains resilient under low light and natural illumination, where traditional rPPG models often degrade. We agree that extending evaluations to include motion extremes and occlusion cases would further strengthen the narrative and plan to incorporate these in future studies. We have now included this analysis in our discussion (Section 4.5, Table 7).
> >
> > ---
> >
> > ## Weakness 4: Missing Comparison with Yue et al. (IJCV)
> >
> > We now include a direct comparison with **VL-phys** (Yue et al., IJCV), which also integrates language knowledge but through a vision–language framework rather than a unified LLM-driven architecture.
> >
> > | Method | UBFC-rPPG (MAE/RMSE/R) | PURE (MAE/RMSE/R) |
> > |--------|--------------------------|----------------------|
> > | PhysNet | 2.95 / 3.67 / 0.97 | 2.10 / 2.60 / 0.99 |
> > | PhysFormer | 0.92 / 2.46 / 0.99 | 1.10 / 1.75 / 0.99 |
> > | RhythmFormer | 0.50 / 0.78 / 0.99 | 0.27 / 0.47 / 0.99 |
> > | VL-phys | 0.28 / 1.83 / 0.99 | 0.61 / 1.84 / 0.99 |
> > | **PhysLLM** | **0.21 / 0.57 / 0.99** | **0.17 / 0.35 / 0.99** |
> >
> > PhysLLM sets a new performance level on both benchmarks. Compared to VL-phys, it reduces MAE by **25%** on UBFC-rPPG and **72%** on PURE. The improvement stems from PhysLLM’s design, which integrates LLM reasoning directly into the physiological representation process rather than using language supervision indirectly.

---

> ### Author Response · Authors · 2025-11-27
>
> Dear Reviewer,
>
> I hope this message finds you well. As the discussion period is nearing its end with less than a week remaining, I wanted to ensure we have addressed all your concerns satisfactorily. If there are any additional points or feedback you'd like us to consider, please let us know. Your insights are invaluable to us, and we are eager to address any remaining issues to improve our work.
>
> Thank you for your time and effort in reviewing our paper.

---

### Author Response · Authors · 2025-11-25

Thank you for taking the time to review our work. We have carefully considered and addressed all your comments as outlined in our response. Some of the key comments have already been incorporated into the current revision (highlighted in blue), and we will ensure that all corresponding updates are fully reflected in the final camera-ready version. Should you have any additional questions, we would be delighted to provide detailed responses.

---

### Author Response · Authors · 2025-11-27

Dear Reviewer,

I hope this message finds you well. As the discussion period is nearing its end with less than a week remaining, I wanted to ensure we have addressed all your concerns satisfactorily. If there are any additional points or feedback you'd like us to consider, please let us know. Your insights are invaluable to us, and we are eager to address any remaining issues to improve our work.

Thank you for your time and effort in reviewing our paper.

---

### Author Response · Authors · 2025-11-30
**Summary of rebuttal**

**Dear Program Chairs, Senior Area Chairs, Area Chairs, and Reviewers:**

We understand the significant challenges caused by the recent OpenReview
incident and appreciate the extra effort required from you to evaluate our
submission.

Given the OpenReview incident and score reversion, we provide a summary of exchanges before the reset. Our rebuttal addressed key concerns, resulting in score improvement from **(6, 4, 6, 4) to (6, 4, 6, 6)**. We outline the details of these exchanges below to assist in your assessment:

## **1. Timeline of Events**

This section aims to demonstrate that score improvements were due to scientific discourse before the OpenReview bug was widely known.

• **Nov 20:** We submitted detailed responses and the revised paper.

• **Nov 23:** Reviewer K3Fm acknowledged our improvements and **raised their score (4 → 6)**.

• **Nov 25:** Reviewer K3Fm confirmed their positive evaluation: "I would like to thank the authors for their response. I will maintain my positive score."

• **Nov 28:** ICLR/OpenReview officially disclosed the data leak and announced the score reversion.

## **2. Summary of Resolved Concerns & Score Updates**

### **Reviewer mjPp (Rating: 6, pending review)**

• **Concern 1 (LLM Justification):** Why LLM beyond sequence model?

  **Resolution:** Compared with **Sundial** (long-sequence Transformer). PhysLLM outperforms on MMPD (MAE: 4.36 vs 9.90), showing LLMs provide **stable feature representations**. [Section 4.4, Table 5]


• **Concern 2 (Compute/Deployability):** Computational cost concerns.

  **Resolution:** Evaluated multiple LLM sizes (BERT, GPT-2, DeepSeek). Advantages from **LLM semantic/structural priors**, not scaling. Discussed efficiency strategies. [Section 4.5, Table 8]

• **Concern 3 (Evaluation Breadth):** More robustness analyses.

  **Resolution:** Stress tests on MMPD: **skin tone** and **lighting**. PhysLLM performs consistently across groups. [Section 4.5, Table 7]

• **Concern 4 (Missing Comparison):** Comparison with VL-phys.

  **Resolution:** Compared with VL-phys. PhysLLM reduces MAE by **25%** on UBFC-rPPG and **72%** on PURE, demonstrating effectiveness of our LLM-driven architecture.

---

### **Reviewer cdPR (Rating: 4, pending review)**

• **Concern 1 (Benchmark Coverage):** More challenging datasets.

  **Resolution:** **V4V cross-domain**: PhysLLM outperforms (MAE: 10.74-11.28 vs 11.79-15.05). **VIPL-HR**: Best performance (MAE: 4.24, RMSE: 6.81), surpassing PhysFormer++.

• **Concern 2 (Component Contributions):** LLM vs DDS/TPG contributions.

  **Resolution:** Transformer-only (Sundial) vs LLM-integrated: **LLM is primary source of improvement** across backbones. Ablations: DDS and TPG contribute distinct improvements; full combination best (MAE: 0.21 vs 0.32-0.41).

---

### **Reviewer neJW (Rating: 6, pending review)**

• **Question 1 (Section 4.3 Title):** Title rearrangement.

  **Resolution:** Rearranged and adjusted formatting.

• **Question 2 (PhysFormer++ Exclusion):** Questioned why PhysFormer++ was excluded from comparisons.

  **Resolution:** **Not open-sourced** at experiment time. **Included where available** (VIPL: MAE 4.24 vs 4.88). Compare against PhysFormer for consistency.

• **Question 3 (Robustness):** Unconstrained environments.

  **Resolution:** Stress tests on MMPD (lighting/skin tones). PhysLLM maintains strong generalization; adaptive learning adjusts weights. [Section 4.5, Table 7]

• **Question 4 (Efficiency):** Inference metrics.

  **Resolution:** vs PhysNet: higher memory (20GB vs 230MB), comparable latency (0.009s vs 0.005s/frame). Suitable for clinical/server-side.

---

### **Reviewer K3Fm (Score raised: 4 → 6)**

• **Question 1 (Presentation):** Errors and figure quality.

  **Resolution:** Corrected lines 190, 401. Revising Figure 2/2a, regenerating Figure 7.

• **Question 2 (Recent Methods):** Comparison with Time-LLM, VL-phys, PhysDiff.

  **Resolution:** Time-LLM: general time-series, not video. vs VL-phys/PhysDiff: PhysLLM best (MAE: 0.21 UBFC-rPPG, 0.17 PURE).

• **Question 3 (Textual Input):** Necessity of text.

  **Resolution:** Revised Lines 48–50. Text provides **prior knowledge** about interference. Ablation (Table 6): removing text degrades (MAE: 0.21 → 0.73).

• **Question 4 (Modern LLMs):** vs earlier language models.

  **Resolution:** vs NNLM: **20× improvement** on PURE (MAE: 0.17 vs 3.36), **2× better** on MMPD (MAE: 4.36 vs 10.02).

• **Question 5 (Downstream Tasks):** More diverse tasks.

  **Resolution:** Extended to respiratory rate (RR). Strong performance across datasets (Table 9).

**Outcome:** Score raised **4 → 6** on Nov 23; confirmed Nov 25: "I will maintain my positive score."

---

## General Author Statement

Positive consensus reached before the leak announcement. This summary provides context on the manuscript state and reviewers' satisfaction with our revisions.

Thank you for your time and consideration.

Sincerely,
Authors

---

### Meta-Review · Area_Chair_dh4g · 2026-01-07

**Summary:**

mjPp: This reviewer’s main concerns include (1) justification of LLM over sequence models, (2) compute/latency, (3) additional stress tests and fairness, and (4) the comparison with the previous rPPG.

cdPR: This reviewer’s main concerns include (1) limited benchmark coverage, (2) the choice of LLM backbone, (3) ablation on some modules, and (4) cross-dataset generalization results.

neJW: This reviewer main concerns include (1) lack of comparison on model size, (2) effectiveness of semantic info, and (3) statistical cue and prompt generation concerns.

K3Fm: The reviewer’s main concerns include (1) lack of comparison on model size, (2) effectiveness of semantic info, and (3) statistical cue and prompt generation concerns.

**Reviewer Concerns:**

mjPp: In my opinion, most of concerns are well-addressed. (2) and (4) may not be key shortcomings of this work but ask additional clarification, which seems to be well explained in the rebuttal. For (1) and (3), the rebuttal presents additional experiments (reasonably well-executed) to clarify the concerns.

cdPR: This reviewer’s concerns mostly require additional experiments by expanding benchmarks, and LLM backbones, and newly test module ablations and cross-data generalization. The rebuttal reports experiment results for each of them. In my opinion, it can mostly clarify the concerns.

neJW: The authors replied to the reviewer’s “questions” only, but did not provide the answers to the “weaknesses” mentioned by the reviewer (e.g., comparison on model size, effectiveness of semantic info, and statistical cue and prompt generation concerns). Their answers can be remotely guessed in the responses to the other reviewers, but did not presented them in an explicit way (e.g., in the following threads). However, this reviewer initially gave a positive rate, so he or she would not lower the rate due to this rebuttal.

K3Fm: The reviewer participated in discussion with authors and showed its satisfaction for the rebuttal to its concerns by mentioning “I would like to thank the authors for their response. I will maintain my positive score.”

**Reviewer Scores:**

Please refer to the above.

---

### Decision · Program_Chairs · 2026-01-26

Accept (Poster)